# LogiNumSynth: Synthesizing Joint Logical-Numerical Reasoning Problems for Language Models

## Abstract

Joint logical-numerical reasoning remains a major challenge for language models, yet existing datasets rely on fixed rule sets and offer limited control over task complexity, constraining their generalizability for evaluation and training. We present *LogiNumSynth*, a flexible natural language problem synthesizer that synthesizes tasks requiring proficiency in joint logical reasoning (e.g., rule-based reasoning) and numerical reasoning (e.g., arithmetic computation). LogiNumSynth supports fine-grained control over reasoning world richness, logical reasoning depth, and the complexity of numerical computations, enabling flexible data synthesis across difficulty levels. We demonstrate three key contributions: (1) *Synthesizer*—synthesizing fully controllable joint reasoning tasks over natural language; (2) *Evaluation & Process Analysis*—evaluating both process accuracy and answer accuracy; (3) *Targeted Training*—using synthesized data to enhance LLMs' reasoning performance. Experiments with multiple LLMs highlight persistent weaknesses in logical-numerical reasoning, showing that LogiNumSynth can serve as both a diagnostic tool and a source of targeted supervision for advancing integrated reasoning skills.

## 1 Introduction

Comprehending the world and adeptly applying knowledge in practical scenarios pose fundamental challenges for natural language processing (NLP). At the core of this cognitive process lie *logical reasoning* and *numerical reasoning*, which jointly enable solving complex problems and deriving meaningful insights (Saxton et al., 2019; Clark et al., 2020; Liu et al., 2020; Hendrycks et al., 2021b).

**Motivation.** Although considerable efforts have been made to advance various aspects of reasoning (Lample & Charton, 2019; Li et al., 2022; Lu et al., 2022; Wei et al., 2022; Lightman et al., 2023a; Li et al., 2023; Xu et al., 2024), existing datasets typically focus on *either* logical reasoning (Clark et al., 2020; Morishita et al., 2023; 2024) *or* numerical reasoning (Cobbe et al., 2021; Chen et al., 2021b; Veeraboina, 2023; Gao et al., 2025) in isolation (see Appendix C for related work). In practice, however, many real-world problems require these abilities to be integrated. Table 1 shows a biology example where solving the task demands both the application of genetic rules (logical reasoning) and probabilistic computation of phenotypic ratios (numerical reasoning).

RuleArena (Zhou et al., 2025) is, to our knowledge, the only existing dataset explicitly integrating logical and numerical reasoning, covering tasks such as taxation, airline luggage fees, and NBA trade validation. However, it has the following limitations. First, it offers limited control over logical and numerical complexity, and its reliance on fixed rules leads to repetitive task patterns, which limit its extensibility and generalizability for evaluation and targeted training. Second, while the original work reports rule-level recall and precision (a coarse form of process evaluation), it does not provide fine-grained assessment of intermediate reasoning steps. These limitations motivate the need for a flexible synthesizer that can (1) synthesize joint logical-numerical problems of adjustable complexity for evaluation and further serving as a targeted training resource to improve reasoning performance, (2) incorporate structured intermediate steps to enable fine-grained process-level diagnosis.

Table 1: Example of joint logical-numerical reasoning in biology.

| Problem | Gene A is dominant over a (A = red, a = white). Dominant B suppresses A, producing white regardless of A/a. Two loci assort independently. Given $AaBb \times AaBb$, determine the offspring phenotypic ratio. |
|---|---|
| Reasoning | *Logical reasoning*: $B\_ \to$ white; *bb*: $A\_ \to$ red, $aa \to$ white. 
 *Numerical reasoning*: From Mendelian segregation, $\Pr(B\_) = \frac{3}{4}$ (all white), $\Pr(bb) = \frac{1}{4}$: among them $\frac{3}{4}$ red, $\frac{1}{4}$ white. Combining: white $= \frac{3}{4} + \frac{1}{4} \times \frac{1}{4} = \frac{13}{16}$; red $= \frac{1}{4} \times \frac{3}{4} = \frac{3}{16}$. Ratio $= 3{:}13$. |

**Our Work.** We introduce *LogiNumSynth*, a controllable natural language problem synthesizer that synthesizes tasks requiring integrated logical reasoning (e.g., rule-based reasoning) and numerical reasoning (e.g., arithmetic computation). Our synthesizer allows *fine-grained control* over reasoning world richness (e.g., the number of rules and facts), logical reasoning depth, and the complexity of numerical computations (e.g., computation range), enabling flexible synthesis of customized datasets across difficulty levels. It is also *highly extensible*: the rule language supports easy inclusion of new mathematical expressions and richer logical operators, allowing the construction of more diverse and challenging reasoning processes tailored to specific evaluation needs. It operates *independently of domain-specific background knowledge*, ensuring that evaluation reflects a model's inherent reasoning capability rather than its memorized factual knowledge.

To synthesize a problem instance, LogiNumSynth first constructs a formal world of automatically synthesized facts and rules, which incorporate numerical expressions, paired with a query, and then translates this structure into natural language descriptions. Models must reason over the provided natural language facts and rules to answer the query. In addition to answer accuracy, we evaluate *process accuracy* by comparing a model's generated reasoning steps with the automatically derived gold-standard reasoning process.

**Our Main Contributions** are:

- *Synthesizer:* A customizable and extensible synthesizer that synthesizes natural language problems requiring joint logical and numerical reasoning with controllable complexity.
- *Evaluation & Process Analysis:* A large-scale evaluation of 29 models, including process-level reasoning diagnostics.
- *Targeted Training:* Demonstrating that our synthetic data can enhance model performance on external reasoning tasks.

**Code and Data** are available in the supplementary material.

## 2 DATA SYNTHESIZATION

Our synthesization starts from synthetic formal representations on which reasoning combines logical and numerical operations. Then we convert formal representations into diverse templated natural language descriptions and use an LLM to improve their fluency. We summarize the features of our synthesizer in Appendix D. Now we elaborate on its design.

### 2.1 FORMAL REPRESENTATION

To synthesize a problem instance (called a *sample* in the rest of the paper) that requires reasoning, we first define its underlying *formal representation* ⟨Facts, Rules, Query⟩. Facts and Rules jointly establish a *world model* where formal reasoning can be executed to derive the gold-standard answer for the given Query. Specifically, facts that match the condition of a rule will trigger this rule to infer new facts. An example is shown in the "Reasoning DAG" part of Figure 1.

**Entities, Attributes, and Relationships.** Each sample is constructed from elements that include a set of *entities* $\mathbf{E} = \{e_1, e_2, \cdots\}$ and a set of *attributes* $\mathbf{A} = \{a_1, a_2, \cdots\}$. Each entity is associated

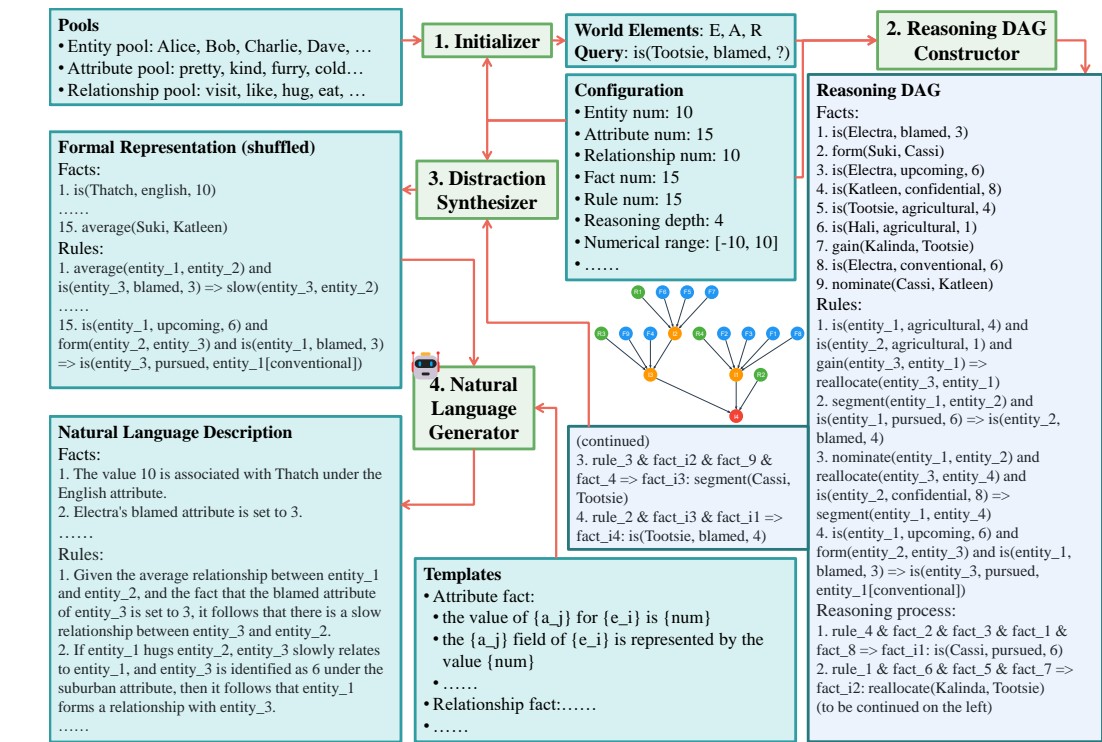

Figure 1: Overview of sample synthesization.

with a number of attributes. The *value* of an attribute is numerical and characterizes the degree of the attribute in practical scenarios. For example, when we declare that the *cold* attribute of the entity *Dave* has a value of 4, it signifies the level or intensity of coldness that he experiences, with 4 indicating a certain degree of severity. We also include binary *relationships* between entities. The set of all relationships is denoted by $\mathbf{R} = \{r_1, r_2, \cdots\}$.

**Facts** include all known values of the attributes of the entities and their relationships. They provide the foundational information needed for reasoning. Facts come in two types:

- $r_k(e_i, e_j)$, denoting the existence of a directed relationship $r_k \in \mathbf{R}$ between two entities $e_i, e_j \in \mathbf{E}$, and

- $\text{is}(e_i, a_j, \text{num})$, which specifies a numerical value num for a specific attribute $a_j \in \mathbf{A}$ of an entity $e_i \in \mathbf{E}$.

**Rules** are expressed in an implication form, composed of a `condition` and a `conclusion`:

$$\text{Condition} \rightarrow \text{Conclusion}.$$

If `condition` is true, then `conclusion` is inferred to be true.

In our current implementation (which can be easily extended), the `condition` is formulated as a conjunction of one or more facts, while the `conclusion` is derived as a corresponding fact. Crucially, all entities within these facts are represented as anonymized placeholder variables, denoted using Greek letters as subscripts (e.g., $e_\alpha$, $e_\beta$). These placeholders are dynamically instantiated with concrete entities during rule application. To ensure logically valid inferences, any placeholder variable appearing in the conclusion must also appear in the corresponding condition. Formally, we define the `condition` using regular expression as:

$$[\text{is}(e_\alpha, a_i, \text{num}) \mid r_j(e_\beta, e_\gamma)] \; \big[ \wedge \; [\text{is}(e_\epsilon, a_k, \text{num}') \mid r_l(e_\zeta, e_\iota)] \big]^*.^1$$

---

[1] i.e., $[\text{is}(e_\alpha, a_i, \text{num}) \mid r_j(e_\beta, e_\gamma)] \wedge \cdots \wedge [\text{is}(e_\epsilon, a_k, \text{num}') \mid r_l(e_\zeta, e_\iota)]$

Table 2: Examples and numbers of templates used for converting formal representation into natural language.

|  | Formal representation | Example of natural language template | #Templates |
|---|---|---|---|
| Attribute fact | $\text{is}(e_i, a_j, \text{num})$ | the value of $\{a_j\}$ for $\{e_i\}$ is $\{\text{num}\}$ | 222 |
| Relationship fact | $r_k(e_i, e_j)$ | $\{e_i\}$ $\{r_k\}$ $\{e_j\}$ | 20 |
| Implication | $condition \rightarrow conclusion$ | whenever $\{condition\}$, $\{conclusion\}$ | 26 |
| Retrieval expression | $e_\alpha[a_i]$ | the value of $\{a_i\}$ for $\{e_\alpha\}$ | 16 |
| Calculation expression (addition) | $k \times e_\alpha[a_i] + b$ | multiplying $\{e_\alpha[a_i]\}$ by $\{k\}$ and adding $\{b\}$ | 15 |
| Calculation expression (subtraction) | $k \times e_\alpha[a_i] - b$ | multiplying $\{e_\alpha[a_i]\}$ by $\{k\}$ and subtracting $\{b\}$ | 15 |
| Aggregation expression (max) | $\max(\text{expr1}, \text{expr2})$ | the greater of $\{\text{expr1}\}$ and $\{\text{expr2}\}$ | 6 |
| Aggregation expression (min) | $\min(\text{expr1}, \text{expr2})$ | the minimum value of $\{\text{expr1}\}$ and $\{\text{expr2}\}$ | 6 |
| Aggregation expression (addition) | $\texttt{addition}(\text{expr1}, \text{expr2})$ | the total of $\{\text{expr1}\}$ and $\{\text{expr2}\}$ | 6 |
| Aggregation expression (subtraction) | $\texttt{subtraction}(\text{expr1}, \text{expr2})$ | the difference between $\{\text{expr1}\}$ and $\{\text{expr2}\}$ | 6 |

Similarly, the `conclusion` is implemented as a factual statement of the form:

- $r_i(e_\alpha, e_\beta)$, or
- $\text{is}(e_\gamma, a_j, \text{expr}(\dots))$, where $\text{expr}(\dots)$ is a function over placeholder variables defined in the `condition`.

Unlike facts, conclusions involving $\text{expr}(\dots)$ introduce numerical computations. In the current implementation, we support four types of expressions:

- **Constant expression:** a parameter-free expression representing a fixed constant value.
- **Retrieval expression:** parameterized by an entity $e_\alpha$ and an attribute $a_i$; its value is simply the attribute of the entity itself, denoted as $e_\alpha[a_i]$.
- **Calculation expression:** parameterized by an entity $e_\alpha$ and an attribute $a_i$; its value is computed as a linear form $k \times e_\alpha[a_i] + b$, where $k$ and $b$ are intrinsic parameters of the expression sampled from a specified range, as detailed in Appendix E.1.
- **Aggregation expression:** defined over two sub-expressions (each being a constant, retrieval, or calculation expression). Its parameters are inherited from the sub-expressions, and its value is obtained by aggregating the results of the two sub-expressions using one of the supported operators: `max`, `min`, `addition`, or `subtraction` (e.g. $\max(k \times e_\alpha[a_i] + b, e_\beta[a_j])$).

**Query** finally asks the model to determine the value of a specific attribute for a given entity and can be formulated as $\text{is}(e_i, a_j, ?)$.

A reasoning model capable of answering such queries requires proficiency in *joint rule-based logical reasoning and arithmetic-based numerical reasoning*. Note that *the above implementation can be easily extended* by, for example, designing more diverse forms of rules and providing more expressive arithmetic operations. These extensions are incremental and are left for future work.

## 2.2 SAMPLE SYNTHESIZATION

We develop an initializer, a reasoning DAG constructor, a distraction synthesizer, and a natural language generator to synthesize samples described in natural language, as illustrated in Figure 1. These four components equip our synthesizing process with diversity, purity (i.e., independent of background knowledge), and extensibility.

**Initializer** randomly selects entities, attributes, and relationships from predefined pools according to the given configurations (i.e., the numbers of entities, attributes, and relationships), thereby constructing the world elements of the sample inference world. The selection process for **A** and **R** explicitly excludes synonyms, thereby mitigating the risk of semantic ambiguities that could confound the results. Specifically, we use an entity pool of 7,944 items from `nltk.corpus.names`, an attribute pool of 1,366 items from `nltk.corpus.treebank`, and a relationship pool of 976 items from `nltk.corpus.wordnet`. These resources are provided by the Natural Language Toolkit (NLTK) (Bird et al., 2009), a widely used library for natural language processing. After that, the initializer randomly selects one entity and one attribute as the query.

**Reasoning DAG Constructor**   takes as input the world elements, the query, and the configuration parameters (e.g., the reasoning depth). It then attempts to construct a reasoning directed acyclic graph (DAG) by working backward from the query as the target node, with each node representing either a fact or a rule along the reasoning path. Specifically, the constructor first takes the query as the target and synthesizes a rule that can derive this target. The number of conditions and the form of the conclusion in this rule are randomly determined according to the configuration parameters. Next, it synthesizes the corresponding facts that satisfy the rule's conditions and treats these facts as the subsequent targets. For each new target, the constructor either directly produces a supporting fact or synthesizes another rule together with additional targets as conditions to support it. This process continues until the reasoning depth specified in the configuration is reached. During the generation of each fact or rule, the constructor ensures that no conflicts occur, i.e., respecting attribute value uniqueness, thus guaranteeing the uniqueness of the query's answer. Finally, it performs forward reasoning across the DAG to derive both the gold-standard reasoning process and final answer. Further implementation details are provided in Appendix E.1.

**Distraction Synthesizer**   produces the remaining distracting facts and rules based on the constructed reasoning DAG and the configuration parameters (e.g., the numbers of facts and rules). Similar to the construction of reasoning DAG, the facts and rules are randomly synthesized according to the configurations, while ensuring that no conflicts arise with the existing facts and rules.

**Natural Language Generator**   operates in two steps. First, templates are used to convert the formal representation of the synthesized sample into diverse, template-based natural language descriptions, laying the foundation for the subsequent natural language optimization. The number of predefined templates is summarized in Table 2. In particular, to describe a rule, we randomly select and combine templates for its components. For example, describing the rule $r_i(e_\alpha, e_\beta) \rightarrow$ $\mathrm{is}(e_\beta, a_j, 3 \times e_\alpha[a_k] - 9)$ requires one template for describing the implication, two templates to describe the two facts, one template to describe the calculation expression, and one template to describe the retrieval expression. In the second step, it takes the template-based descriptions and further refines them using an LLM, making the text more fluent and natural. This refinement ensures that the generated text is syntactically accurate and semantically coherent, making it suitable for processing by both humans and language models. More examples of templates and detailed settings are provided in Appendix E.2.

## 3    EXPERIMENT SETUP

We synthesized datasets of varying difficulty (detailed in Section 3.1) to evaluate language models of different architectures and scales (detailed in Section 3.2), with evaluation metrics described in Section 3.3, as well as to serve as training resources (detailed in Section 3.4).

### 3.1    SYNTHETIC DATASETS

We first synthesized four datasets using our data synthesizer, named EL-EN, EL-HN, HL-EN and HL-HN (each containing 500 samples). The first part of the name represents the logical reasoning difficulty (Easy or Hard Logical), while the second part indicates the numerical reasoning difficulty (Easy or Hard Numerical). To further assess the reasoning capabilities of state-of-the-art models, we also synthesized exHL-HN (extremely Hard Logical-Hard Numerical), containing 400 samples. Lastly, for the training resources for reasoning, we synthesized two datasets: EN-Train and EL-Train (each containing 5000 samples). These are specifically designed to lower the difficulty in numerical (Easy Numerical) or logical (Easy Logical) reasoning, respectively, to aid model training toward the other reasoning capability. We controlled the difficulty of the samples in each dataset by varying factors such as reasoning depth, the number of conditions in a logical rule, the range of numerical computations, and the types of expressions used. More details on the synthesizing configurations of these datasets are provided in Appendix E.3, and we further discuss the quality of the synthesized data in Appendix E.4.

## 3.2 TESTED MODELS

We evaluated 29 LLMs, including models with long reasoning chains: GPT5-mini (based on GPT5-mini-2025-08-07) (OpenAI, 2025), DeepSeek-R1 (based on DeepSeek-R1-0528) (Guo et al., 2025); models specialized for reasoning: Phi4-mini-reasoning (Xu et al., 2025), Phi4-reasoning and Phi4-reasoning-plus (Abdin et al., 2025); hybrid reasoning models: Qwen3 series (0.6B, 1.7B, 4B, 8B, 14B, 32B) (Yang et al., 2025); models with MoE (Mixture of Experts) architectures: Qwen3-30B-A3B (Yang et al., 2025); and common instruct models: Llama3.1-8B-instruct (Dubey et al., 2024), Llama3.2-1B-instruct, Llama3.2-3B-instruct (AI, 2024), Qwen2.5-instruct series (0.5B, 1.5B, 3B, 7B, 14B, 32B) (Team, 2024), GLM4-chat (GLM et al., 2024), GLM4-airx, -0520, -plus[2], Phi4-mini-instruct (Abouelenin et al., 2025), Phi4 (Abdin et al., 2024), DeepSeek-V3 (based on DeepSeek-V3-0324) (Liu et al., 2024), and GPT-4o (based on GPT-4o-2024-11-20) (OpenAI, 2023), with parameter sizes ranging from 0.5B to 671B.

Large models with long reasoning chains and significant parameter sizes, such as GPT5-mini and DeepSeek-R1, demonstrate the current upper bound of reasoning capabilities in LLMs. On the other hand, models with moderate parameter sizes, such as Qwen3-14B and Phi4-reasoning-plus (14B), represent a balanced scale, offering a trade-off between computational efficiency and reasoning capability. Smaller models, such as Qwen3-1.7B and Llama3.2-1B-instruct, further emphasize computational efficiency, but with a reduction in reasoning depth and complexity. We prompted them using Chain of Thought (CoT), in the zero-shot and few-shot settings (Wei et al., 2022). Implementation details are provided in Appendix F.1.

## 3.3 EVALUATION METRICS

Due to the nature of our synthesized datasets, each question's reasoning can be decomposed into multiple steps, enabling a fine-grained evaluation of a model's *process accuracy*—the extent to which its reasoning steps match the gold-standard process—and *answer accuracy*—the correctness of the final result. To compute process accuracy, we prompted models to explicitly summarize their reasoning at the end and used the structured output mode of Qwen3-8B to extract structured reasoning steps from their output. Both the raw and structured outputs were compared with the gold-standard reasoning process, allowing partial credit for intermediate steps that were correct. In cases where the model outputs a different reasoning path but still arrives at the correct final answer, the process is considered correct. This step-level evaluation is particularly important in scenarios such as automated theorem proving, multi-step medical diagnosis, and financial auditing, where the transparency, rigor, and correctness of each step are as critical as the final result. The implementation of process accuracy is detailed in Appendix F.2.

## 3.4 TRAINING SETUP

To demonstrate the potential of our synthetic data as an additional training resource for enhancing model reasoning ability, we conducted supervised fine-tuning on Qwen3-1.7B and Llama3.2-1B-instruct using LoRA (Hu et al., 2021), and, in some experiments, the Recall Adam Optimizer Chen et al. (2020). The models were evaluated on numerical reasoning benchmarks: GSM8K (Cobbe et al., 2021), MATH (Hendrycks et al., 2021b), MATHQA (Amini et al., 2019), SVAMP (Patel et al., 2021), MAWPS (Koncel-Kedziorski et al., 2016), AIME (Veeraboina, 2023); formal deductive logical reasoning benchmarks: RuleTaker (Clark et al., 2020), ProofWriter (Tafjord et al., 2020), FOLIO (Han et al., 2024a), FLD (Morishita et al., 2023); complex logical, joint logical-numerical reasoning, and common benchmarks: LogiQA (Liu et al., 2020), ReClor (Yu et al., 2020), AbductionR (Young et al., 2022), RuleArena (Zhou et al., 2025), MMLU (Hendrycks et al., 2021a), CLUTRR (Sinha et al., 2019), SLR-Bench Helff et al. (2025), ProntoQA (Saparov & He, 2023). Detailed training configurations are provided in Appendix F.3.

---

[2]https://open.bigmodel.cn/

Table 3: Performance of LLMs on datasets of different difficulty levels. 'Proc' refers to process accuracy and 'Ans' refers to answer accuracy.

| Model | #Params. | #Shots | EL-EN | | EL-HN | | HL-EN | | HL-HN | | Average | |
|---|---|---|---|---|---|---|---|---|---|---|---|---|
| | | | Proc | Ans | Proc | Ans | Proc | Ans | Proc | Ans | Proc | Ans |
| Llama3.2-1B-instruct | 1.24B | 0-shot | 0.07 | 14.20 | 0.00 | 1.40 | 1.34 | 9.20 | 0.00 | 1.20 | 0.35 | 6.50 |
| | | 3-shot | 0.50 | 15.00 | 0.00 | 1.80 | 0.00 | 7.80 | 0.00 | 0.40 | 0.12 | 6.25 |
| Llama3.2-3B-instruct | 3.21B | 0-shot | 4.07 | 33.20 | 0.10 | 11.60 | 0.20 | 14.00 | 0.00 | 3.00 | 1.09 | 15.45 |
| | | 3-shot | 1.57 | 27.80 | 0.40 | 4.60 | 0.47 | 20.40 | 0.00 | 4.00 | 0.61 | 14.20 |
| Llama3.1-8B-instruct | 8.03B | 0-shot | 20.87 | 50.20 | 5.90 | 16.00 | 2.31 | 34.00 | 0.83 | 5.00 | 7.48 | 26.30 |
| | | 3-shot | 12.60 | 48.60 | 2.97 | 13.20 | 0.29 | 34.60 | 0.17 | 5.20 | 4.01 | 25.40 |
| Qwen2.5-0.5B-instruct | 494M | 0-shot | 0.40 | 6.20 | 0.00 | 0.40 | 0.20 | 4.80 | 0.00 | 0.40 | 0.15 | 2.95 |
| | | 3-shot | 0.00 | 7.80 | 0.00 | 0.00 | 0.00 | 3.80 | 0.00 | 0.60 | 0.00 | 3.05 |
| Qwen2.5-1.5B-instruct | 1.54B | 0-shot | 0.40 | 7.40 | 0.00 | 0.20 | 1.20 | 4.60 | 0.00 | 0.00 | 0.40 | 3.05 |
| | | 3-shot | 0.60 | 14.20 | 0.00 | 0.60 | 0.00 | 6.00 | 0.00 | 0.20 | 0.15 | 5.25 |
| Qwen2.5-3B-instruct | 3.09B | 0-shot | 0.80 | 12.40 | 0.00 | 1.20 | 1.60 | 9.20 | 0.00 | 0.60 | 0.60 | 5.85 |
| | | 3-shot | 1.00 | 19.20 | 0.00 | 0.80 | 0.20 | 6.60 | 0.00 | 0.00 | 0.30 | 6.65 |
| Qwen2.5-7B-instruct | 7.62B | 0-shot | 20.40 | 56.00 | 9.80 | 33.40 | 1.28 | 29.40 | 1.31 | 15.60 | 8.20 | 33.60 |
| | | 3-shot | 9.17 | 45.40 | 2.97 | 15.20 | 0.42 | 33.40 | 0.07 | 5.60 | 3.16 | 24.90 |
| Qwen2.5-14B-instruct | 14.8B | 0-shot | 48.40 | 83.60 | 36.27 | 68.40 | 4.05 | 59.40 | 3.47 | 36.20 | 23.04 | 61.90 |
| | | 3-shot | 29.00 | 71.80 | 8.83 | 24.00 | 2.97 | 58.00 | 2.14 | 15.00 | 10.73 | 42.20 |
| Qwen2.5-32B-instruct | 32.8B | 0-shot | 60.37 | 85.40 | 12.43 | 22.00 | 0.20 | 4.80 | 0.05 | 2.60 | 18.26 | 28.70 |
| | | 3-shot | 36.43 | 78.80 | 16.73 | 35.80 | 0.66 | 6.80 | 0.52 | 1.20 | 13.59 | 30.65 |
| Qwen3-0.6B | 752M | 0-shot | 11.67 | 43.20 | 0.80 | 21.80 | 0.00 | 24.20 | 0.00 | 5.60 | 3.12 | 23.70 |
| | | 3-shot | 12.70 | 47.20 | 6.33 | 27.00 | 0.05 | 21.20 | 0.16 | 8.40 | 4.81 | 25.95 |
| Qwen3-1.7B | 2.03B | 0-shot | 12.97 | 76.80 | 4.47 | 62.20 | 2.30 | 19.20 | 0.89 | 6.80 | 5.16 | 50.45 |
| | | 3-shot | 31.53 | 80.60 | 22.50 | 64.20 | 1.39 | 45.80 | 0.82 | 20.40 | 14.06 | 52.75 |
| Qwen3-4B | 4.02B | 0-shot | 56.83 | 90.60 | 39.20 | 78.80 | 2.51 | 37.00 | 1.25 | 11.00 | 24.95 | 54.35 |
| | | 3-shot | 58.70 | 88.20 | 53.27 | 79.80 | 2.26 | 22.60 | 0.57 | 6.80 | 28.70 | 49.35 |
| Qwen3-8B | 8.19B | 0-shot | 69.30 | 90.80 | 27.80 | 60.80 | 5.05 | 31.40 | 4.48 | 11.40 | 26.66 | 48.60 |
| | | 3-shot | 65.67 | 87.40 | 56.37 | 79.80 | 2.55 | 23.00 | 1.64 | 4.80 | 31.56 | 48.75 |
| Qwen3-14B | 14.8B | 0-shot | 30.40 | 69.80 | 24.13 | 54.60 | 15.01 | 54.60 | 16.45 | 54.60 | 21.50 | 59.80 |
| | | 3-shot | 43.50 | 54.80 | 34.13 | 42.20 | 7.15 | 53.20 | 9.59 | 61.00 | 23.59 | 52.80 |
| Qwen3-30B-A3B | 30.5B | 0-shot | 51.27 | 66.40 | 51.00 | 66.40 | 18.22 | 77.40 | 15.28 | 75.00 | 33.94 | 71.30 |
| | | 3-shot | 46.33 | 59.40 | 43.73 | 57.60 | 15.77 | 72.80 | 17.03 | 74.60 | 30.72 | 66.10 |
| Qwen3-32B | 32.8B | 0-shot | 75.97 | 91.60 | 68.43 | 87.00 | 14.96 | 81.20 | 13.49 | 72.20 | 43.21 | 83.00 |
| | | 3-shot | 78.50 | 84.80 | 72.00 | 79.00 | 15.59 | 81.40 | 15.06 | 74.00 | 45.29 | 79.80 |
| GLM4-9b-chat | 9.4B | 0-shot | 0.40 | 4.40 | 0.00 | 0.20 | 0.20 | 3.80 | 0.00 | 0.00 | 0.15 | 2.10 |
| | | 3-shot | 0.13 | 9.00 | 0.00 | 1.20 | 0.00 | 3.60 | 0.00 | 0.20 | 0.03 | 3.50 |
| GLM4-airx | >10B | 0-shot | 37.13 | 63.80 | 25.43 | 43.00 | 2.38 | 24.20 | 0.24 | 5.80 | 16.30 | 34.20 |
| | | 3-shot | 31.77 | 64.60 | 12.50 | 28.00 | 1.38 | 39.80 | 0.56 | 8.20 | 11.55 | 35.15 |
| GLM4-0520 | >100B | 0-shot | 45.50 | 68.80 | 27.83 | 40.00 | 1.03 | 27.40 | 0.40 | 6.20 | 18.69 | 35.60 |
| | | 3-shot | 42.27 | 61.20 | 25.57 | 37.80 | 1.56 | 28.00 | 0.35 | 6.40 | 17.43 | 33.35 |
| GLM4-plus | >100B | 0-shot | 79.43 | 93.00 | 59.50 | 68.60 | 19.53 | 69.20 | 8.96 | 36.00 | 41.86 | 66.70 |
| | | 3-shot | 75.90 | 93.80 | 54.50 | 61.60 | 14.09 | 69.80 | 7.61 | 30.80 | 38.02 | 64.00 |
| Phi4-mini-instruct | 3.84B | 0-shot | 17.97 | 52.20 | 7.90 | 33.40 | 1.60 | 17.20 | 0.49 | 7.60 | 6.99 | 27.60 |
| | | 3-shot | 6.20 | 38.40 | 0.00 | 0.20 | 0.00 | 4.41 | 0.00 | 0.00 | 1.55 | 10.75 |
| Phi4-mini-reasoning | 3.84B | 0-shot | 4.80 | 65.20 | 1.17 | 50.20 | 2.38 | 40.00 | 1.47 | 20.60 | 2.46 | 44.00 |
| | | 3-shot | 5.60 | 63.60 | 1.47 | 49.60 | 3.24 | 43.20 | 1.36 | 21.40 | 2.92 | 44.45 |
| Phi4 | 14.7B | 0-shot | 65.63 | 89.60 | 57.73 | 76.40 | 2.31 | 36.20 | 2.01 | 20.80 | 31.92 | 55.75 |
| | | 3-shot | 63.10 | 90.20 | 51.17 | 74.40 | 2.52 | 36.20 | 2.06 | 10.80 | 29.71 | 52.90 |
| Phi4-reasoning | 14.7B | 0-shot | 0.90 | 22.60 | 0.20 | 6.40 | 0.00 | 10.20 | 0.07 | 5.40 | 0.29 | 11.95 |
| | | 3-shot | 0.30 | 24.20 | 0.17 | 8.20 | 0.44 | 10.00 | 0.08 | 2.20 | 0.25 | 11.15 |
| Phi4-reasoning-plus | 14.7B | 0-shot | 63.87 | 97.60 | 57.73 | 89.60 | 27.44 | 83.40 | 20.56 | 65.40 | 42.40 | 84.00 |
| | | 3-shot | 64.17 | 97.60 | 57.90 | 89.80 | 27.75 | 83.60 | 20.47 | 65.40 | 42.57 | 84.10 |
| DeepSeek-V3 | 685B | 0-shot | 90.07 | 98.40 | 81.80 | 89.60 | 43.97 | 89.60 | 30.93 | 68.00 | 61.69 | 86.40 |
| | | 3-shot | 88.37 | 95.20 | 74.77 | 84.20 | 25.56 | 79.20 | 19.87 | 52.40 | 52.14 | 77.75 |
| DeepSeek-R1 | 685B | 0-shot | 97.87 | 99.00 | 91.93 | 92.60 | 86.77 | 98.20 | 76.35 | 92.60 | 88.23 | 95.60 |
| | | 3-shot | 97.27 | 99.80 | 92.33 | 91.80 | 83.73 | 98.20 | 75.39 | 92.80 | 87.18 | 95.65 |
| GPT-4o | - | 0-shot | 77.43 | 92.00 | 59.97 | 79.00 | 9.60 | 59.40 | 5.12 | 38.80 | 38.03 | 67.30 |
| | | 3-shot | 76.73 | 90.20 | 64.33 | 76.20 | 14.55 | 68.60 | 8.60 | 43.40 | 41.05 | 69.60 |
| GPT5-mini | - | 0-shot | 98.40 | 99.60 | 92.93 | 92.00 | 92.50 | 99.40 | 79.72 | 94.40 | 90.89 | 96.35 |
| | | 3-shot | 98.87 | 100.00 | 93.50 | 92.80 | 91.66 | 99.80 | 79.72 | 94.80 | 90.94 | 96.85 |

Table 4: Performance of two top-performing models on the exHL-HN dataset in the 0-shot setting. 'Proc' refers to process accuracy and 'Ans' refers to answer accuracy.

| Model | EL-EN | | EL-HN | | HL-EN | | HL-HN | | exHL-HN | |
|---|---|---|---|---|---|---|---|---|---|---|
| | Proc | Ans | Proc | Ans | Proc | Ans | Proc | Ans | Proc | Ans |
| DeepSeek-R1 | 97.87 | 99.00 | 91.93 | 92.60 | 86.77 | 98.20 | 76.35 | 92.60 | 12.27 | 38.50 |
| GPT5-mini | 98.40 | 99.60 | 92.93 | 92.00 | 92.50 | 99.40 | 79.72 | 94.40 | 23.23 | 54.75 |

# 4 EXPERIMENT RESULTS

## 4.1 EVALUATION RESULTS

**Most Models Perform Poorly.** The evaluation results of LLMs are shown in Table 3. Due to the inherent difficulty of the synthesized tasks, many commonly used models perform poorly. For instance, Llama3.1-8B-instruct achieves only 0.83 process accuracy and 5.00 answer accuracy in the 0-shot setting on HL-HN, while GPT-4o scores 5.12 and 38.80, respectively. In contrast, stronger reasoning models like DeepSeek-R1 and GPT5-mini achieve 76.35 and 92.60, and 79.72 and 94.40, respectively.

**Analysis of Two Best Models.** For GPT5-mini and DeepSeek-R1, transferring from EN to HN results in a noticeable decrease in both process accuracy and answer accuracy, consistent with other models. However, transferring from EL to HL causes a more significant drop in process accuracy (e.g. from 91.93 on EL-HN to 76.35 on HL-HN for DeepSeek-R1 in the 0-shot setting), while answer accuracy remains relatively unchanged (e.g. from 92.60 on EL-HN to 92.60 on HL-HN for DeepSeek-R1 in the 0-shot setting). This phenomenon is observed only in these two models. This suggests that these two models are generally capable of handling the logical reasoning complexity of the tasks to produce correct answers, but struggle with presenting clear and logically sound reasoning processes.

Given the strong reasoning abilities demonstrated by the two top-performing models, we further stress-tested them using exHL-HN in the 0-shot setting. As shown in Table 4, they fail to perform well (e.g., GPT5-mini only scores 23.23 and 54.75 as process and answer accuracy respectively), and the results highlight GPT5-mini's stronger reasoning ability compared to DeepSeek-R1. This difference is less apparent in the previous four datasets, showcasing how our synthesizer can synthesize datasets of varying difficulty levels to evaluate the capabilities of different models.

**EL-HN v.s. HL-EN Comparison.** Figure 2a shows the process accuracy and answer accuracy of different models on EL-HN and HL-EN, indicating that while HL slightly increases the difficulty for answer accuracy, models perform significantly worse on process accuracy under HL, compared with HN. This is due to the larger reasoning depth and more complex logical rules, which significantly hinder the models' ability to provide accurate reasoning processes.

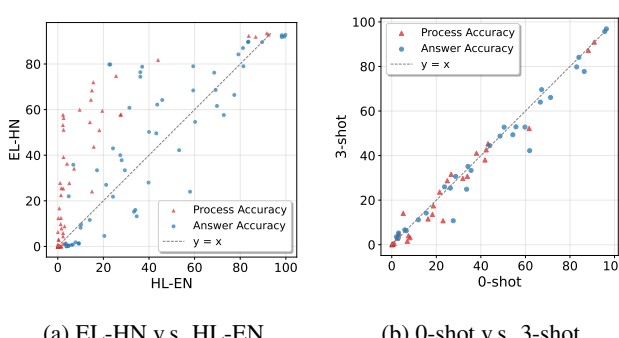

(a) EL-HN v.s. HL-EN      (b) 0-shot v.s. 3-shot

Figure 2: Analyses of evaluation results.

**0-shot v.s. 3-shot Comparison.** Figure 2b shows the comparison of average accuracy across different datasets for 29 models in the 0-shot and 3-shot settings, indicating that, across all models, 3-shot learning generally has little effect, with some models showing improvement and others experiencing a decline in performance. This is mainly because the synthesized tasks are inherently complex reasoning problems. While the provided examples can help guide the model on how to approach the problem, they may also divert the model's attention, leading to performance changes. We also conducted experiments with varying numbers of few-shot examples, as detailed in Appendix G.1, and the results remain consistent.

Table 5: Performance on external reasoning benchmarks before and after SFT on our synthetic data. For benchmark-specific settings, hyperparameters are tuned individually for each benchmark, and for Llama3.1-8B-instruct we additionally select between EL-Train and EN-Train using the corresponding validation set. In contrast, the unified training row ( + LogiNumSynth (unified training) ) uses a single set of hyperparameters tuned once and applied to all benchmarks. An asterisk (*) denotes results reported by Helff et al. (2025).

| Model | GSM8k | MATH | MATHQA | SVAMP | MAWPS | AIME | Average |
|---|---|---|---|---|---|---|---|
| Llama3.2-1B-instruct | 35.50 | 27.33 | 22.23 | 61.00 | 74.72 | 2.85 | 37.27 |
| + LogiNumSynth (EL-Train) | 37.37 (+1.87) | 28.63 (+1.30) | 23.00 (+0.77) | 61.66 (+0.66) | 75.78 (+1.60) | 4.22 (+1.37) | 38.44 (+1.17) |
| + LogiNumSynth (unified training) | 44.35 (+8.85) | 28.06 (+0.73) | 25.80 (+3.57) | 61.00 | 75.30 (+0.58) | 3.06 (+0.21) | 39.60 (+2.33) |
| Llama3.1-8B-instruct | 83.40 | 40.05 | 50.90 | 85.66 | 93.90 | 9.93 | 60.64 |
| + LogiNumSynth | 83.78 (+0.38) | 39.71 (-0.34) | 54.81 (+3.91) | 87.67 (+2.01) | 94.19 (+0.29) | 10.14 (+0.21) | 61.72 (+1.08) |
| Qwen3-1.7B | 88.77 | 67.16 | 66.52 | 94.33 | 93.99 | 24.49 | 72.54 |
| + LogiNumSynth (EL-Train) | 89.23 (+0.46) | 67.77 (+0.61) | 66.80 (+0.28) | 95.33 (+1.00) | 94.52 (+0.53) | 25.55 (+1.06) | 73.20 (+0.66) |

| Model | RuleTaker | ProofWriter | FOLIO | FLD | Average |
|---|---|---|---|---|---|
| Llama3.2-1B-instruct | 46.18 | 24.84 | 35.47 | 32.60 | 34.77 |
| + LogiNumSynth (EN-Train) | 46.60 (+0.42) | 25.80 (+0.96) | 35.96 (+0.49) | 32.90 (+0.30) | 35.32 (+0.54) |
| + LogiNumSynth (unified training) | 46.60 (+0.42) | 24.90 (+0.06) | 35.47 | 31.90 (-0.70) | 34.72 (-0.05) |
| Llama3.1-8B-instruct | 57.30 | 27.70 | 39.90 | 31.10 | 39.00 |
| + LogiNumSynth | 58.10 (+0.80) | 28.80 (+1.10) | 43.84 (+3.94) | 42.36 (+11.26) | 43.28 (+4.28) |
| Qwen3-1.7B | 70.29 | 75.30 | 52.36 | 60.17 | 64.53 |
| + LogiNumSynth (EN-Train) | 71.90 (+1.61) | 79.50 (+4.20) | 62.56 (+10.20) | 63.90 (+3.73) | 69.47 (+4.94) |

| Model | LogiQA | ReClor | AbductionR | RuleArena | MMLU | CLUTRR | SLR-Bench | ProntoQA |
|---|---|---|---|---|---|---|---|---|
| Llama3.2-1B-instruct | 12.85 | 28.43 | 47.30 | 1.96 | 37.95 | 7.63 | 0.00 | 47.00 |
| + LogiNumSynth (EN-Train) | 23.35 (+10.50) | 30.20 (+1.77) | 51.80 (+4.50) | 2.62 (+0.66) | 38.03 (+0.08) | 8.78 (+1.15) | 0.00 | 49.60 (+2.60) |
| + LogiNumSynth (unified training) | 15.53 (+2.68) | 24.90 (-3.53) | 48.10 (+0.80) | 0.84 (-1.12) | 21.05 (-16.90) | 5.82 (-1.81) | 0.00 | 51.63 (+4.63) |
| Llama3.1-8B-instruct | 30.10* | 50.00 | 48.40 | 10.80 | 63.30* | 17.55 | 22.3 | 90.60 |
| + LogiNumSynth | 53.74 (+23.64) | 51.80 (+1.80) | 69.10 (+20.7) | 11.42 (+0.62) | 64.31 (+1.01) | 13.74 (-3.81) | 22.7 (+0.40) | 91.60 (+1.00) |
| Qwen3-1.7B | 56.41 | 31.80 | 40.80 | 8.18 | 64.68 | 13.65 | 7.50 | 94.00 |
| + LogiNumSynth (EN-Train) | 58.09 (+1.68) | 35.80 (+4.00) | 43.00 (+2.20) | 10.30 (+2.12) | 67.28 (+2.60) | 18.03 (+4.38) | 7.70 (+0.20) | 97.00 (+3.00) |

**Case Study and Error Analysis.** We conducted a case study that reveals several interesting observations. We found instances where the answer is wrong but the process accuracy is non-zero, demonstrating that process accuracy can capture partial correctness in reasoning even when the final answer fails. We also observed a few rare cases where a model discovers a shorter yet correct reasoning path that differs from the gold-standard reasoning path. This is possible and considered correct because our synthesizer guarantees logical consistency (i.e., no derived facts, including the query, contradict each other) while allowing the existence of multiple valid reasoning paths. For cases where the answer accuracy equals 1 but the process accuracy is below 1, the errors in the reasoning process typically fall into three categories: (1) **incorrect application of rules**: misapplication or omission of relevant facts or rules, yet resulting in a correct intermediate conclusion; (2) **incorrect numerical computations**: errors in intermediate numerical calculations, e.g., the final answer is calculated as $\max(a, b)$ where $a$ is incorrect but $b$ is correct and larger; (3) **incorrect intermediate results**: producing wrong intermediate conclusions—commonly caused by lexical errors in entities, attributes and relationships or by reversing the direction of relationships—but ultimately yielding the correct final answer. Representative examples are provided in Appendix G.2.

## 4.2 TRAINING RESULTS

To further assess the utility of our synthetic data as an *additional training resource* for improving reasoning capabilities, we fine-tuned some models on the synthesized datasets and directly evaluated them on existing reasoning benchmarks. We consider two complementary settings: (1) benchmark-specific adaptation and (2) unified training on a mixed synthetic corpus.

**Benchmark-specific adaptation.** In the first setting, we fine-tuned models separately for each benchmark. Hyperparameters were tuned independently for every benchmark using the corresponding validation set, and for Llama3.1-8B-instruct we additionally tuned the choice between EL-Train and EN-Train per benchmark. Benchmark-specific results for all three models are reported in Table 5; for Llama3.2-1B-instruct, this table also includes the unified training variant described below. We discuss the rationale for this benchmark-specific fine-tuning design in Appendix B.1.

We first present the pre- and post-fine-tuning performance of Llama3.2-1B-instruct, Llama3.1-8B-instruct and Qwen3-1.7B across multiple out-of-domain numerical and formal deductive logical reasoning benchmarks, along with their average scores. Results show consistent improvements across all tasks, ranging from relatively simple datasets such as GSM8k to more challenging benchmarks like AIME. Notably, Qwen3-1.7B achieves a remarkable +10.20 improvement on FOLIO, while Llama3.1-8B-instruct gains +23.64 on LogiQA.

We also report results on datasets that involve more diverse reasoning types—LogiQA (diverse logical reasoning and reading comprehension), AbductionR (abductive logical reasoning), RuleArena (rule-guided numerical reasoning), MMLU (common knowledge QA), CLUTRR and SLR-Bench (inductive logic reasoning), etc. These results illustrate the potential of our synthetic data to serve as an effective training resource for enhancing reasoning abilities in out-of-domain settings.

**Unified training on mixed EN-Train and EL-Train.**   In the second setting, we additionally train a single general-purpose model on the combination of EL-Train and EN-Train with a fixed configuration and evaluate it across all benchmarks. We conduct this unified training experiment for Llama3.2-1B-instruct, whose results are reported in Table 5 in the row labeled `+ LogiNumSynth (unified training)`. Compared to the benchmark-specific setting, this unified configuration yields clear gains on several reasoning-focused benchmarks such as GSM8k and LogiQA, but also leads to noticeable trade-offs on others, most prominently on the broad-coverage knowledge QA benchmark MMLU.

## 5    CONCLUSION

Recent NLP research has increasingly focused on equipping language models with robust logical and numerical reasoning capabilities. To contribute to this direction, we introduced *LogiNumSynth*, a data synthesizer designed to benchmark models' integrated logical-numerical reasoning abilities. With four synthetic datasets of varying difficulty levels, we evaluated 29 models covering diverse architectures and scales, and conducted a detailed analysis of their joint reasoning performance. In addition, we synthesized a more challenging dataset to assess the strongest two models, revealing that there remains substantial room for improvement. Beyond evaluation, LogiNumSynth also provides synthetic training resources that can effectively enhance models' reasoning abilities.

The extensibility of LogiNumSynth makes it a *living benchmark*: researchers can easily extend it to new domains and reasoning settings by incorporating additional operations or logical constructs, thereby enabling continuous and targeted evaluation of ever-advancing large language models. We provide additional discussion and a detailed examination of the limitations of our work in Appendix B.

### REPRODUCIBILITY STATEMENT

To facilitate the reproducibility of our work, we provide a detailed description of the experimental settings in Section 3, Appendix E.2, Appendix E.3, and Appendix F. In addition, our code and datasets are included in the supplementary material to enable independent verification of our results.

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

## A  THE USE OF LARGE LANGUAGE MODELS

We used large language models to polish the writing.

## B    DISCUSSION AND LIMITATIONS

### B.1    DISCUSSION

**Implicit Assumptions.**    Our framework rests on two key assumptions: (1) that joint logical-numerical reasoning can be effectively modeled using rule-based logical structures combined with numerical expressions, and (2) that expressing these reasoning tasks in natural language preserves their core computational challenges. We further assume that such integrated tasks capture a meaningful subset of real-world reasoning difficulties encountered by LLMs.

**Expectations.**    As outlined in Section 1, LogiNumSynth was developed with two primary expectations: (1) that its controllable joint reasoning tasks would systematically expose LLM weaknesses in intermediate reasoning steps; and (2) that fine-tuning on these tasks would lead to improvements not only on joint reasoning benchmarks, but also on isolated logical or numerical tasks. Our empirical results provide support for both of these expectations.

**Rationale for Benchmark-Specific Fine-Tuning.**    This split training setup is motivated by both realistic usage and the nature of LogiNumSynth. First, in many real applications, task- or scenario-specific adaptation of smaller models is more practical than deploying a single universal large model: practitioners often fine-tune a compact backbone for a target domain or benchmark, rather than expecting one general model to perform optimally everywhere.

Second, this focus follows from the properties of LogiNumSynth itself: it is a controllable synthesis framework, not a fixed distribution, where operator sets and composition depth can be instantiated to match different reasoning regimes. A central question we study is how benchmark-aligned configurations improve the corresponding tasks, which naturally leads to benchmark-specific tuning experiments.

Third, our training uses LoRA-based fine-tuning: instead of retraining separate full models, we attach lightweight LoRA adapters to a shared backbone. This closely mirrors realistic workflows in which small adapters are added for narrow reasoning domains, making the split setting closer to practical deployment.

### B.2    LIMITATIONS

**Better process evaluation.**    Although we take care to minimize errors in process evaluation, it remains an open challenge (Lightman et al., 2023b; Zheng et al., 2024; Lu et al., 2024; Liu et al., 2025b; Zhou et al., 2025). Current process evaluation methods are still imperfect, and a small number of such errors are inevitable, especially when intermediate reasoning steps are lengthy, interleaved, or involve intricate cross-references that disrupt the logical flow.

**Training for more models and benchmarks.**    Due to limitations in computational resources, we only experimented with fine-tuning Llama3.2-1B-instruct, Llama3.1-8B-instruct, and Qwen3-1.7B using our synthetic data, and evaluated them primarily on logical and numerical reasoning benchmarks. Extending to a wider range of models, including larger-scale LLMs and diverse architectures, as well as more benchmarks such as standard general-purpose ones, would further validate the generality of our synthesizer and potentially yield more substantial improvements in reasoning performance.

**Data-related limitations.**    First, the relationships in our synthesized worlds (e.g., "$a$ visits $b$") do not carry real-world semantic meaning, a limitation also common in prior work on logical synthetic datasets (Clark et al., 2020; Tafjord et al., 2020; Morishita et al., 2023; 2024). Currently, there is no effective solution to this issue other than relying on expert annotation (Han et al., 2024a;b). To mitigate ambiguity, we avoid introducing synonyms within the same sample, though deeper semantic grounding remains an open challenge. Second, our current synthesizer includes only conjunction and implication as logical structures, and four basic types of numerical expressions. While the synthesizer is inherently extensible, incorporating more complex logical constructs and numerical operations is left to future work. Third, LogiNumSynth only covers a subset of real-world reasoning scenarios and does not incorporate tasks requiring extensive domain knowledge.

**Theoretical Limitations.** While synthetic fine-tuning enhances performance, it does not resolve the fundamental theoretical limitations of current LLMs—an important direction for future research that falls outside the scope of this work.

## C  RELATED WORK

### C.1  LOGICAL AND NUMERICAL REASONING DATASETS

Logical reasoning and numerical reasoning represent two of the most fundamental reasoning types in NLP (Wang et al., 2022; Yu et al., 2024), and considerable effort has been made in logical reasoning (Clark et al., 2020; Li et al., 2022) and numerical reasoning (Wei et al., 2022; Li et al., 2023).

Datasets involving a single type of reasoning, such as logical reasoning, including Ruletaker (Clark et al., 2020), ProofWriter (Tafjord et al., 2020), ReClor (Yu et al., 2020), AR-LSAT (Zhong et al., 2021), FOLIO (Han et al., 2024a), PFOLIO (Han et al., 2024b), LOGIGLUE (Luo et al., 2023), FLD (Morishita et al., 2023), FLDx2 (Morishita et al., 2024), LogicNLI (Tian et al., 2021), and more recently SLR (Helff et al., 2025). SLR focuses on symbolic inductive reasoning over a finite domain with human-defined grammars; however, its "numeric" symbols do not support arithmetic operations or quantitative constraints, and its evaluations remain within purely logical benchmarks (e.g., LogiQA, LogiQA2), without numerical reasoning.

Similarly, several numerical reasoning datasets such as MAWPS (Koncel-Kedziorski et al., 2016), DROP (Dua et al., 2019), MathQA (Amini et al., 2019), GSM8k (Cobbe et al., 2021), FinQA (Chen et al., 2021b), SVAMP (Patel et al., 2021), MATH (Hendrycks et al., 2021b), AIME (Veeraboina, 2023), and Omni-MATH (Gao et al., 2025) require identifying numerical structures and performing computations, but do not couple them with logical constraints.

A few datasets attempt to explore broader forms of reasoning. KOR-Bench(Ma et al., 2025) evaluates knowledge-orthogonal reasoning through separate logical and numerical tasks; however, it does not include tasks requiring the joint integration of logical structure with numeric computation, and is positioned solely as an evaluation suite. SynLogic(Liu et al., 2025a) synthesizes 35 diverse logical task types (e.g., Sudoku-like puzzles) using symbolic constraint systems, but these tasks primarily involve structural or relational logic and largely lack explicit arithmetic or quantitative reasoning requirements.

*A notable exception is RuleArena* (Zhou et al., 2025), which, to the best of our knowledge, is the only existing benchmark that explicitly combines logical and numerical reasoning in natural language. It includes three real-world inspired tasks (taxation, airline luggage fee calculation, and NBA transaction validation) that require applying logical rules together with numerical computations. While RuleArena makes an important contribution by integrating these two forms of reasoning, it adopts a fixed set of rules and scenarios with limited linguistic variation, resulting in repetitive task patterns and offering little control over the logical and numerical complexity of problem instances. This limits its suitability for probing models at varying levels of difficulty or for covering a broader diversity of reasoning scenarios. Furthermore, although the original work reports rule-level recall and precision (providing a coarse form of process evaluation), it does not perform fine-grained, step-by-step assessment of intermediate reasoning stages. Finally, RuleArena is positioned purely as an evaluation benchmark, with no intended support for targeted training data generation, which further constrains its extensibility.

These limitations motivate the development of more *flexible* and *scalable* synthesis frameworks. Our proposed LogiNumSynth addresses these gaps by supporting precise control over reasoning complexity, synthesizing richer and more varied natural language formulations, and incorporating structured intermediate reasoning steps for fine-grained process-level diagnosis. The dual applicability of LogiNumSynth for evaluation and targeted training enables systematic improvement of models' joint logical-numerical reasoning capabilities.

## C.2 BENCHMARKING REASONING CAPABILITIES

RuleTaker (Clark et al., 2020) evaluated RoBERTa (Liu et al., 2019) using synthetic logical reasoning data. Based on this, FLD (Morishita et al., 2023) and LogicNLI (Tian et al., 2021) extended the expressiveness of logical reasoning by introducing more deduction rules and evaluated more models. There are also works that evaluated LLMs such as ChatGPT (Brown et al., 2020) and GPT-4 (OpenAI, 2023) using logical reasoning problems from standard exams (Liu et al., 2023; 2020; Yu et al., 2020; Wang et al., 2022). HumanEval (Chen et al., 2021a), MMLU (Hendrycks et al., 2021a), BIG-bench (Srivastava et al., 2022), and AIME (Veeraboina, 2023) have become standard benchmarks to evaluate the foundational language and reasoning capabilities of language models.

In contrast, we *incorporate joint logical-numerical reasoning while minimizing confounding factors such as background knowledge*, to ensure that the evaluation results more purely reflect models' reasoning capabilities. Furthermore, we conduct an evaluation of 29 models, with parameters from 0.5B to 685B, to observe the differences in their reasoning capabilities. In addition, unlike existing synthetic datasets (Clark et al., 2020; Morishita et al., 2023; Tian et al., 2021; Morishita et al., 2023; 2024), which used a limited set of templates for natural language generation, we *employ hundreds of templates and an LLM to generate diverse, syntactically accurate, and semantically coherent natural language descriptions* for problem instances.

## D FEATURES OF THE DATA SYNTHESIZER

Our data synthesizer exhibits diversity, purity, and extensibility, as described below.

### D.1 DIVERSITY

Our data synthesizer achieves diversity through controllable scale and difficulty, specifically in world richness, depth of reasoning, and arithmetic computation.

**World Richness** has two aspects: scale and density. Scale describes the size of the world through counts of entities, attributes, and relationships. Density reflects the complexity of interconnections among entities' attribute values and relationships, as established by the numbers of facts and rules. In worlds with higher density, these elements are more tightly intertwined, enabling richer and more extensive inference chains. By varying scale and density, our synthesizer can produce knowledge bases ranging from simple, sparsely connected worlds to highly interwoven reasoning environments.

**Depth of Reasoning.** The difficulty of reasoning can be controlled by specifying the required depth of logical reasoning.

**Arithmetic Computation.** The difficulty of arithmetic computation is tuned through types of expressions (from constant expression to nested aggregation expression), their sampling distribution ratios, and ranges of operands.

### D.2 PURITY

Unlike existing datasets derived from real applications which essentially evaluate a hybrid of reasoning capability and knowledge capacity of a reasoning model, we seek to decouple reasoning from knowledge so that our evaluation can be focused on reasoning capability. This is reflected in our synthesizing process, where world elements are combined randomly so that the knowledge stored in a reasoning model (e.g., commonsense knowledge learned during pre-training) is unlikely to help. Only the facts and rules provided should be useful. It helps to support less biased evaluation and offer a clearer view of the reasoning performance of models, representing an aspect that is pivotal for advances in areas where reasoning is prioritized over knowledge retention and retrieval.

### D.3 EXTENSIBILITY

Our data synthesizer has a high degree of extensibility. In particular, the expressions in the rules are designed to be extensible. As mentioned in Section 2.1, we currently incorporated only four

---

**Algorithm 1** Reasoning DAG Construction (Part 1/2)

---

**Require:**
1: World elements $\mathbf{E}, \mathbf{A}, \mathbf{R}$
2: Query $q$
3: Configuration $config$
**Ensure:**
4: DAG $\mathcal{G}$
5: Facts `Facts`
6: Rules `Rules`
7: **function** CONSTRUCTDAG($\mathbf{E}, \mathbf{A}, \mathbf{R}, q, config$)
8:     $state_0 \leftarrow (\emptyset, \emptyset, \emptyset, \emptyset)$           ▷ Initialize empty state: $(\mathcal{G}, \texttt{Facts}, \texttt{Rules}, \mathcal{C})$
9:     $state_{final} \leftarrow$ BUILDNODE($q, 0, state_0, config$)     ▷ Build DAG starting from query
10:    Extract $(\mathcal{G}, \texttt{Facts}, \texttt{Rules}, \mathcal{C})$ from $state_{final}$     ▷ Extract final components
11:    **return** $\mathcal{G}, \texttt{Facts}, \texttt{Rules}$
12: **end function**

---

13: **function** BUILDNODE($node, depth, state, config$)
14:    Extract $(\mathcal{G}, \texttt{Facts}, \texttt{Rules}, \mathcal{C})$ from $state$
15:    **if** $depth = config.depth$ **then**          ▷ At maximum depth, must synthesize fact
16:       **return** SYNTHESIZEFACT($node, state$)
17:    **else**
18:       **if** currently last node to synthesize **then**   ▷ Not at required depth, must synthesize rule for last node
19:          **return** SYNTHESIZERULE($node, depth, state, config$)
20:       **else**
21:          $p \leftarrow$ random($[0, 1]$)          ▷ Roll dice for synthesis strategy
22:          **if** $p < 0.5$ **then**
23:             **return** SYNTHESIZERULE($node, depth, state, config$)
24:          **else**
25:             **return** SYNTHESIZEFACT($node, state$)
26:          **end if**
27:       **end if**
28:    **end if**
29: **end function**

---

types of expressions. However, our synthesizer is adaptable to the inclusion of additional mathematical expressions such as exponentiation and trigonometric functions. We recommend selecting different expressions based on the specific requirements to evaluate the model's capabilities in the corresponding scenarios.

Also, as mentioned in Section 2.1, our rule condition is a conjunction of one or more atoms. It can be extended to support richer logical operators (e.g., disjunctions $\lor$ and negations $\neg$), thereby inducing more complex reasoning process.

# E   DETAILS ON DATA SYNTHESIZATION

We present the method for synthesizing formal representations in Appendix E.1, additional templates and details in Appendix E.2, the configuration settings for dataset synthesis in Appendix E.3, and quality discussion of synthesized data in Appendix E.4.

## E.1   DETAILS ON FORMAL REPRESENTATION SYNTHESIZATION

The sample synthesization process outlined in Section 2.2 consists of four sequential stages. Following the initialization of world elements and query, the subsequent two stages construct the formal representation $\langle \texttt{Facts}, \texttt{Rules}, \texttt{Query} \rangle$ as defined in Section 2.1, which will be illustrated in the following. The final stage generates corresponding natural language descriptions, with implementation details provided in Section 2.2 and Appendix E.2.

---

**Algorithm 2** Reasoning DAG Construction (Part 2/2)

---

30: **function** SYNTHESIZEFACT($node, state$)
31:     Extract $(\mathcal{G}, \texttt{Facts}, \texttt{Rules}, \mathcal{C})$ from $state$
32:     **repeat**                                      ▷ Generate facts until finding a valid one
33:         Generate fact $f$ for $node$
34:         $\mathcal{C}_{temp} \leftarrow$ UPDATECONCLUSIONS($\mathcal{C}, \texttt{Facts} \cup \{f\}, \texttt{Rules}$)
35:         $valid \leftarrow$ (no conflict between $f$ and $\mathcal{C}_{temp}$)
36:     **until** $valid$
37:     $\texttt{Facts}' \leftarrow \texttt{Facts} \cup \{f\}$
38:     Update $\mathcal{G}$ with $f$ to get $\mathcal{G}'$
39:     $\mathcal{C}' \leftarrow \mathcal{C}_{temp}$
40:     **return** $(\mathcal{G}', \texttt{Facts}', \texttt{Rules}, \mathcal{C}')$
41: **end function**

---

42: **function** SYNTHESIZERULE($node, depth, state, config$)
43:     Extract $(\mathcal{G}, \texttt{Facts}, \texttt{Rules}, \mathcal{C})$ from $state$
44:     **repeat**                                      ▷ Generate rules until finding a valid one
45:         Generate rule $r$ for $node$
46:         $\mathcal{C}_{temp} \leftarrow$ UPDATECONCLUSIONS($\mathcal{C}, \texttt{Facts}, \texttt{Rules} \cup \{r\}$)
47:         $valid \leftarrow$ (no conflict in $\mathcal{C}_{temp}$ and no cycle with $r$ in $\mathcal{G}$)
48:     **until** $valid$
49:     $\texttt{Rules}' \leftarrow \texttt{Rules} \cup \{r\}$
50:     Update $\mathcal{G}$ with $r$ to get $\mathcal{G}'$
51:     $\mathcal{C}' \leftarrow \mathcal{C}_{temp}$
52:     $final\_state \leftarrow (\mathcal{G}', \texttt{Facts}, \texttt{Rules}', \mathcal{C}')$
53:     **for** each atom $a$ required in $r$'s condition **do**          ▷ Build prerequisite atoms recursively
54:         $final\_state \leftarrow$ BUILDNODE($a, depth + 1, final\_state, config$)
55:     **end for**
56:     **return** $final\_state$
57: **end function**

---

58: **function** UPDATECONCLUSIONS($\mathcal{C}, \texttt{Facts}, \texttt{Rules}$)
59:     $\mathcal{C}_{current} \leftarrow \mathcal{C} \cup \texttt{Facts}$              ▷ Start with existing conclusions and facts
60:     **repeat**                                      ▷ Apply rules until fixed point is reached
61:         $\mathcal{C}_{old} \leftarrow \mathcal{C}_{current}$
62:         **for** each rule $r$ in $\texttt{Rules}$ **do**                 ▷ Try to apply each rule
63:             **if** $r$ can be triggered by $\mathcal{C}_{current}$ **then**
64:                 Derive new conclusion $c$ from $r$
65:                 $\mathcal{C}_{current} \leftarrow \mathcal{C}_{current} \cup \{c\}$
66:             **end if**
67:         **end for**
68:     **until** $\mathcal{C}_{current} = \mathcal{C}_{old}$
69:     **return** $\mathcal{C}_{current}$
70: **end function**

---

**Overall DAG Construction.** Starting from world elements, query and configurations, we first construct a reasoning directed acyclic graph (DAG) to represent the reasoning process required to answer the query. The DAG is built in a backward, top-down, and recursive manner. The root node is the query, and each non-leaf node represents an intermediate conclusion that can be inferred by applying a rule to its child nodes. Each leaf node corresponds to a fact or a rule that can be directly used to infer its parent node. The depth of the reasoning process is controlled by specified configuration, and the DAG is constructed until the desired depth is reached.

**Rule Synthesis.** To synthesize a rule for inferring an intermediate conclusion or the query, we first randomly decide the number of atoms in the rule condition, which is sampled from a range predefined by configuration. Each atom can be either an attribute fact or a relationship fact, where the entities are anonymized as a variable and can be instantiated with concrete entities during rule

application. For an attribute fact, we randomly select an attribute from the attribute set $\mathbf{A}$ of world elements. For a relationship fact, we randomly select a relationship from the relationship set $\mathbf{R}$. The conclusion of the rule is anonymized by the target intermediate conclusion or the query, so that it can be inferred when the rule is applied. Specially, for an attribute fact in the rule conclusion, an expression is needed to compute the attribute value.

To synthesize an expression, we first randomly select the type from the supported expression types with probabilities predefined by configuration, and then sample the parameters or other components required by the selected expression type. For calculation expressions, we randomly select the parameters $k$ and $b$ from the ranges specified in the configuration. For aggregation expressions, we uniformly select the aggregation operator from the supported set and recursively synthesize the two sub-expressions based on the predefined probabilities of expression types in the configuration.

It is important to ensure that the anonymized entities in the rule conclusion also appear in the rule condition, to avoid introducing new entities during rule application.

**Recursive Node Building.** After synthesizing a rule to infer an intermediate conclusion or the query, more intermediate conclusions may be needed to infer the atoms in the rule condition. We recursively synthesize rules or direct facts for these intermediate conclusions until the desired depth is reached. The decision to synthesize a rule or a fact is made uniformly at random, except at the maximum depth (where a fact must be synthesized) or for the last node at a given level (where a rule is required to ensure progression toward the desired depth).

The algorithm for constructing the reasoning DAG is shown in Algorithm 1 and Algorithm 2. The main procedure begins with the CONSTRUCTDAG function, which initializes an empty state comprising the DAG $\mathcal{G}$, facts `Facts`, rules `Rules`, and inferable conclusions $\mathcal{C}$. It then recursively builds the DAG starting from the query node via BUILDNODE. In BUILDNODE, the synthesis strategy depends on the current depth: at the maximum depth, a fact is synthesized; otherwise, a rule is mandatory for the last node at that level to extend the depth, while for others, the choice between rule and fact is randomized with equal probability.

**Conflict Detection and Validation.** Every time we synthesize a rule or a fact, we ensure that it does not lead to multiple, inconsistent derivations when combined with the existing rules and facts, thereby maintaining uniqueness of the inferred results. Specifically, we maintain a set of all conclusions that can be inferred by the existing rules and facts, and ensure that the new rule or fact, along with its triggered conclusions, does not conflict with them. Additionally, we also ensure that the reasoning DAG remains acyclic, i.e., no intermediate conclusion needs to be used to infer itself. If a conflict is detected or a cycle is introduced, we discard the newly synthesized rule or fact and re-synthesize it until no conflict or cycle is found.

The SYNTHESIZEFACT function generates a fact for the node and iteratively checks for conflicts by temporarily updating the conclusions using UPDATECONCLUSIONS. Once a valid fact is found, it updates the state accordingly. Similarly, SYNTHESIZERULE generates a rule, verifies it for conflicts and cycles, and upon validation, recursively calls BUILDNODE for each atom in the rule's condition to build prerequisite substructures.

**Termination of UPDATECONCLUSIONS.** UPDATECONCLUSIONS computes the iterative closure of a monotone consequence operator $T : 2^{\mathcal{U}} \to 2^{\mathcal{U}}$, where one step updates the current conclusion set $\mathcal{C}$ by $T(\mathcal{C}) = \mathcal{C} \cup \mathrm{Infer}(\mathcal{C}, \texttt{Rules})$. The universe $\mathcal{U}$ consists of (i) attribute assignments $\mathrm{is}(e, a, v)$ and (ii) binary relations $r(e_i, e_j)$. We impose the restriction that for any fixed entity-attribute pair $(e, a)$, at most one value $v$ is admissible. Given finite sets of entities $\mathbf{E}$, attributes $\mathbf{A}$, relations $\mathbf{R}$, and numeric values $\mathbb{V}$ (bounded by current facts and rule parameters), $\mathcal{U}$ is finite, with $\mathcal{U} = \{\mathrm{is}(e, a, v) \mid e \in \mathbf{E}, a \in \mathbf{A}, v \in \mathbb{V}\} \cup \{r(e_i, e_j) \mid r \in \mathbf{R}, e_i, e_j \in \mathbf{E}\}$ and cardinality at most $|\mathbf{E}||\mathbf{A}| + |\mathbf{R}||\mathbf{E}|^2$ for binary $r$. Since $(2^{\mathcal{U}}, \subseteq)$ is a finite lattice and $T$ is monotone, the ascending chain $\mathcal{C}_0 \subseteq \mathcal{C}_1 \subseteq \cdots$ must stabilize after finitely many steps, yielding the least fixed point reachable from the initial state. Conflicts are detected during closure and rejected, ensuring the result is both comprehensive for conflict checks and consistent.

**Distraction Synthesization.** We synthesize additional, potentially irrelevant, facts or rules to distract the reasoning process. These are randomly generated and added to the number specified in

the configuration, while ensuring they do not conflict with existing knowledge. This increases the complexity of the knowledge base, making it harder for models to locate and use the information needed to answer the query.

## E.2 DETAILS ON NATURAL LANGUAGE GENERATOR

Example templates used in the first step of natural language generator for different formal representations are as follows:

1. Attribute fact (i.e. is($e_i, a_j, num$)):
   - the value of $\{a_j\}$ for $\{e_i\}$ is $\{num\}$.
   - $\{num\}$ is recorded for $\{e_i\}$ in the $\{a_j\}$ attribute.
   - the $\{a_j\}$ property for $\{e_i\}$ is given by $\{num\}$.
   - the $\{a_j\}$ field of $\{e_i\}$ is represented by $\{num\}$.

2. Relationship fact (i.e., $r_k(e_i, e_j)$):
   - it can be said that $\{e_i\}$ $\{r_k\}$ $\{e_j\}$.
   - the $\{r_k\}$ correlation is present between $\{e_i\}$ and $\{e_j\}$.
   - in the context of $\{r_k\}$, $\{e_i\}$ and $\{e_j\}$ share a connection.
   - the relationship $\{r_k\}$ defines a connection between $\{e_i\}$ and $\{e_j\}$.

3. Implication (i.e., $condition \rightarrow conclusion$):
   - $\{conclusion\}$ can be deduced from $\{condition\}$.
   - $\{conclusion\}$ is a natural consequence of $\{condition\}$ being true.
   - given $\{condition\}$, $\{conclusion\}$ follows.
   - if $\{condition\}$, $\{conclusion\}$.

4. Retrieval expression (i.e., $e_\alpha[a_i]$):
   - the value of $\{a_i\}$ for $\{e_\alpha\}$.
   - property $\{a_i\}$ of $\{e_\alpha\}$.
   - the value associated with the attribute $\{a_i\}$ of $\{e_\alpha\}$.
   - the value corresponding to $\{a_i\}$ within $\{e_\alpha\}$.

5. Calculation expression (addition, i.e., $k \times e_\alpha[a_i] + b$):
   - multiplying $\{e_\alpha[a_i]\}$ by {k} and adding $\{b\}$.
   - multiplying $\{e_\alpha[a_i]\}$ by {k} with an addition of $\{b\}$.
   - scaling $\{e_\alpha[a_i]\}$ by {k}, then adding $\{b\}$.
   - adding $\{b\}$ to {k} times $\{e_\alpha[a_i]\}$.

6. Aggregation expression (max, i.e., $\max(expr1, expr2)$):
   - the larger of {expr1} and {expr2}.
   - the maximum value between {expr1} and {expr2}.
   - the greater of {expr1} and {expr2}.
   - the maximum of {expr1} and {expr2}.

In the second step, we used Qwen3-8B with thinking mode disabled, running on NVIDIA RTX 3090 and RTX 4090 GPUs with the vLLM framework (Kwon et al., 2023) and PyTorch 2.x. The sampling parameters were: temperature = 0.7, top p = 0.8, and top k = 2. The prompt template and few-shot examples are shown in Table 6.

## E.3 DETAIL ON SYNTHESIZATION CONFIGURATIONS

We synthesized 7 datasets for distinct purposes; their detailed configurations are summarized in Table 7. The first four (EL-EN, EL-HN, HL-EN, HL-HN) target the evaluation of general model reasoning. The exHL-HN dataset is designed to stress-test the best two LLMs: GPT5-mini and DeepSeek-R1; for this dataset, we explicitly synthesized four subsets with reasoning depths from 7 to 10, each containing 100 samples, rather than sampling depths uniformly. The last two (EL-Train, EN-Train) are training resources for strengthening reasoning capabilities of smaller models.

Table 6: Prompt template and examples we used in the second step of natural language generator.

**Prompt.** Given a formal representation in logical form, optimize the corresponding template representation into a fluent, grammatically correct natural language expression that clearly conveys the meaning of the formal representation. Keep the following guidelines in mind:

1. When transforming the template into natural language, ensure that the key terms and relationships (such as specific attributes of entities and the nature of the relationships between them) are preserved exactly. Do not alter critical terms like attribute names, relationships, or the order in which relationships appear.

2. For rule-based formal representations, make sure the causal inference is clearly expressed in the template, highlighting the cause-effect relationship where appropriate.

3. You don't need to strictly follow the template of the examples. Prioritize fluent and diverse language expression, and simply provide the optimized sentence as your answer.

Examples:

{examples}

formal representation: {formal representation}

template representation: {template-based natural language description}

output:

---

**Examples for facts refinement.**

formal representation: is(Susana, low, -8)

template representation: The low of Susana is recorded as -8.

output: Susana's low attribute is recorded as -8.

formal representation: is(Lynn, under, 10)

template representation: The under property for Lynn is represented by the value 10.

output: The under property for Lynn is represented by the value 10.

formal representation: sacrifice(Cecilla, Terrianne)

template representation: Cecilla sacrifice Terrianne.

output: Cecilla sacrifices Terrianne.

formal representation: flood(Jorie, Woodrow)

template representation: It can be said that Jorie flood Woodrow.

output: It can be said that Jorie flooded Woodrow.

---

**Examples for rules refinement.**

formal representation: defuse(entity_1, entity_2) => is(entity_2, technical, subtraction(3 * entity_2[retained] + 8, 5 * entity_2[proven] + 2))

template representation: The difference between multiplying the characteristic retained of entity_2 by 3 with a subtraction of 8 and increasing entity_2's proven 5 times, followed by adding 2 is associated with entity_2 for the technical attribute is a natural consequence of there exists a relationship defuse between entity_1 and entity_2 being true.

output: The relationship between entity_1 and entity_2, denoted as "defuse," implies that entity_2's technical attribute is the result of subtracting the value of five times entity_2's proven characteristic plus 2 from three times entity_2's retained characteristic plus 8.

formal representation: is(entity_1, asian, -1) and is(entity_1, entire, -10) and revise(entity_2, entity_3) => block(entity_3, entity_1)

template representation: Given the asian of entity_1 is -1 and the value -10 is logged for entity_1 under the entire attribute and the revise link is observed between entity_2 and entity_3, entity_3 and entity_1 form a connection of the block relationship follows.

output: Considering that the value -1 is recorded for entity_1 under the asian attribute, and -10 is logged for entity_1 under the entire attribute, along with the existence of a revise relationship between entity_2 and entity_3, it follows that a block relationship is formed between entity_3 and entity_1.

formal representation: is(entity_1, rental, -9) => is(entity_1, monthly, min(5 * entity_1[interstate], 1 * entity_1[deep]))

template representation: The minimum value of scaling the property denoted by interstate for entity_1 by 5 and 1 times the value corresponding to deep within entity_1 is ascribed to entity_1 for the monthly attribute can be safely inferred from within entity_1, the rental attribute is noted as -9.

output: Given that the rental attribute for entity_1 is recorded as -9, it can be inferred that the monthly attribute for entity_1 is assigned the minimum value between 5 times the value of interstate and the value of deep within entity_1 times 1.

formal representation: shop(entity_1, entity_2) and is(entity_3, operating, -6) => is(entity_2, foreign, 0 * entity_2[weakened])

template representation: Given the relationship shop defines a connection between entity_1 and entity_2 and entity_3 is described by -6 within the operating context, the value decreasing the property denoted by weakened for entity_2 by 0 times is ascribed to entity_2 for the foreign attribute follows.

output: Given that the shop relationship exists between entity_1 and entity_2, and entity_3 is described as -6 within the operating context, it follows that the foreign attribute for entity_2 is assigned the value of 0 times the weakened property of entity_2.

The sizes of the world element sets $\mathbf{E}, \mathbf{A}, \mathbf{R}$ are governed by #Entities, #Attributes, and #Relationships. The distraction module expands the knowledge base until the specified #Facts and #Rules are reached. World richness (scale and density) is thus jointly shaped by world sizes and the counts of facts and rules. The Reasoning Depth parameter defines the sampling range for the target inference depth that the DAG constructor is required to realize. #Condition gives either a fixed number or a uniform range for the atoms in a rule body (e.g., [2,3] means sample uniformly from $\{2, 3\}$). Expression Weights define the sampling distribution over expression types (constant, retrieval, calculation, aggregation). For example, "0 0 1 1" permits only calculation and aggregation, equally likely. Aggregation Weights analogously define the distribution of sub-expression types (constant, retrieval, calculation) inside an aggregation (e.g., "1 1 1" = uniform). The intrinsic parameters of calculation expressions ($k, b$ in $expr(e_\alpha, a_i) = k \times e_\alpha[a_i] + b$) and the attribute value ranges in facts are controlled by Oprand Range. For instance, "[-100, 100]" means uniform sampling from $\{-100, -99, \ldots, 99, 100\}$; negative $b$ implicitly yields a subtraction form. Finally, Size is the number of synthesized samples in the dataset.

Our synthesizer offers independent control over structure, interaction density, reasoning depth, rule complexity, numeric difficulty, and dataset size, which supports precise diagnosis during evaluation and serves as a targeted training resource to enhance reasoning capabilities.

### E.4 QUALITY OF SYNTHESIZED DATA

As described in Section 2, the synthesis process uses programmatically controlled synthesization with constrained stochasticity to produce formal task specifications. Each synthesized instance is verified for correctness at the formal level before being converted into natural language, ensuring that every task is logically sound and semantically coherent.

To further evaluate the quality of the generated natural language (NL) descriptions, we specifically aim to verify whether the NL descriptions accurately and fluently reflect their underlying formal representations. Instead of relying on human annotators—which can introduce author-side biases in judgment—we adopt an LLM-as-judge protocol using GPT-4o-2024-11-20. For each fact or rule, the model rates the pair consisting of the formal specification and its NL description along two dimensions: Faithfulness and Fluency.

- **Faithfulness (1-5)**: How accurately the NL text preserves the formal meaning, especially whether the formal specification can be reconstructed from it.

- **Fluency (1-5)**: Grammar, clarity, and naturalness of the NL text.

We apply this evaluation to 6,012 fact pairs and 6,000 rule pairs. The averaged scores are:

- fact faithfulness: 4.73

- fact fluency: 4.91

- rule faithfulness: 4.81

- rule fluency: 4.99

These results indicate that the NL descriptions are both highly faithful to the underlying formal specifications and very fluent. The complete per-instance scores and JSON outputs from the LLM-as-judge evaluation, along with the prompt we used, are provided in the supplementary material under the /quality_scores directory.

Table 7: Detailed configurations of the synthetic datasets in our experiments.

| Dataset | #Entities | #Attributes | #Relationships | #Facts | #Rules | Reasoning Depth | #Condition | Expression Weights | Aggregation Weights | Oprand Range | Size |
|---|---|---|---|---|---|---|---|---|---|---|---|
| EL-EN | 10 | 15 | 10 | 15 | 15 | [1, 3] | 1 | 1 1 1 0 | 1 1 1 | [1, 10] | 500 |
| EL-HN | 10 | 15 | 10 | 15 | 15 | [1, 3] | 1 | 0 0 1 1 | 1 1 1 | [-100, 100] | 500 |
| HL-EN | 10 | 15 | 10 | 15 | 15 | [4, 6] | [2, 3] | 1 1 1 0 | 1 1 1 | [1, 10] | 500 |
| HL-HN | 10 | 15 | 10 | 15 | 15 | [4, 6] | [2, 3] | 0 0 1 1 | 1 1 1 | [-100, 100] | 500 |
| exHL-HN | 30 | 40 | 40 | 15 × Depth | 5 × Depth | [7, 10] | [3, 6] | 0 0 1 1 | 1 1 1 | [-100, 100] | 400 |
| EL-Train | 10 | 15 | 10 | 15 | 15 | [1, 3] | 1 | 1 1 2 2 | 1 1 2 | [-100, 100] | 5000 |
| EN-Train | 10 | 15 | 10 | 15 | 15 | [4, 6] | [2, 3] | 1 1 1 0 | 1 1 1 | [1, 10] | 5000 |

Table 8: Prompt template we used for evaluating LLMs.

# Task
Analyze a logical scenario with entities, their attributes, and relationships. Use the given facts and rules to answer the query through step-by-step reasoning.
## Key Components
- **Entities**: Objects in the scenario
- **Attributes**: Properties of entities (with specific values)
- **Relationships**: Asymmetric connections between entities (direction matters)
- **Facts**: Given information about attributes and relationships
- **Rules**: If-then statements for logical deduction
- **Query**: Question to answer
## Instructions
1. **Natural Analysis**: First, think through the problem freely using clear, natural language. Explain your reasoning process, identify relevant entities, attributes, and relationships, and work toward the solution.
2. **Final Summary**: After your analysis, provide a structured reasoning summary that shows the key logical steps that lead to the answer.
3. **Answer Format**: End with "Answer:
boxed{[value]}"
## Final Summary Requirements
Your summary should list only the complete reasoning steps in this format:
```
[dependencies] => int_[n]: [conclusion]
```
**For relationships**: `rule_X & fact_Y & int_Z => int_n: [relation] exists between [A] and [B]`
**For attributes**: `rule_X & fact_Y & fact_Z & int_W => int_n: [Entity]'s [attribute] is [final_value]`
### Critical Requirements for AttributeFacts:
- Must show the **final calculated value** (e.g., "is 22"), not intermediate expressions
- Must list **ALL dependencies**: the triggering rule and conditions + all facts/intermediates that provide values for the calculation
### Summary Example
```
Reasoning:
rule_15 & fact_13 & fact_4 => int_1: reject exists between Sterne and Beilul
rule_5 & fact_1 & fact_10 & fact_11 & fact_5 => int_2: Nils's prior is 22
rule_8 & int_1 & int_2 & fact_7 => int_3: final_entity's target_attribute is 15
...
Answer: \boxed{37}
```
{examples}
{querying sample: facts, rules, assertion}

# F    IMPLEMENTATION DETAILS

We summarize additional implementation details, including the evaluation settings (Appendix F.1), the process accuracy implementation (Appendix F.2), and the training settings (Appendix F.3).

## F.1    EVALUATION SETTINGS

We accessed the GPT series (GPT-4o and GPT5-mini), DeepSeek series (DeepSeek-V3 and -R1), and three models from the GLM4 series (GLM4-airx, -0520, and -plus) via API calls with default hyperparameters. For GPT5-mini, the reasoning effort is set to the default, i.e. medium. For other LLMs, experiments were conducted on NVIDIA RTX 3090 and 4090 GPUs using vLLM (Kwon et al., 2023) and PyTorch 2.x. The sampling parameters were set to temperature = 0.3, top p = 0.8, and top k = 20. The prompt used is listed in Table 8, and the few-shot examples, synthesized along with the corresponding datasets, can be found in the supplementary material.

Table 9: Prompt template for extracting structured reasoning outputs, where {attributes_list} and {relations_list} denote the attribute set **A** and relationship set **R**, respectively, provided as guidance for the extraction.

---

You are a logical reasoning assistant. Your task is to analyze the given reasoning process and reformat it into a specific structured format.
Requirements:
1. After completing your analysis, summarize the key reasoning steps in the specified structured format
2. The structured format is only required at the end as a summary - your main explanation can be in natural language
3. For the final answer, always use: "Answer: \boxed{[value]}"
Structured Summary Requirements:
"Reasoning:
rule_15 & fact_13 & fact_4 => int_1: relation_name exists between first_entity and second_entity.
rule_5 & fact_1 & fact_10 & fact_11 & fact_5 => int_2: entity_name's attribute_name is attribute_value.
...
Answer: \boxed{answer_value}"
Format Guidelines:
- Each reasoning step should be expressed as: [rule/fact combinations] => int_[n]: [intermediate conclusion]
- Express relationships as "[relation] exists between [X] and [Y]"
- Express attributes as "[X]'s [attribute] is [value]"
- Use logical operators: & (and)
- Number intermediate conclusions sequentially (int_1, int_2, etc.)
Please analyze the following reasoning process and reformat it into the structured format specified above.
Original Answer:
{raw output of llm}
Available Attributes: {attributes_list}
Available Relations: {relations_list}

---

## F.2 IMPLEMENTATION OF PROCESS ACCURACY

Our data synthesizer synthesizes tasks whose reasoning processes can naturally be represented as a reasoning DAG, where each node denotes a known fact, a rule, or an intermediate conclusion (also treated as a fact). As shown in Figure 1, a non-leaf node is derived jointly from all the nodes pointing to it. With the gold-standard reasoning steps produced during synthesis, our dataset is inherently suitable for evaluating a model's reasoning process and gaining deeper insights.

To achieve this, we adopt a dual-extraction approach that combines a text-based parser with an LLM-based structured extractor. Both components operate in parallel and cross-reference each other to accurately reconstruct the reasoning process in a structured form. For the LLM-based extraction, we employ vLLM (Kwon et al., 2023)'s structured output mode, which enforces JSON-formatted outputs during decoding. Specifically, we use Qwen3-8B with the temperature set to 0 and the thinking mode disabled. The prompt used for this structured extraction is provided in Table 9.

After extracting the structured reasoning process of the tested model, we verify each reasoning step by checking: (1) whether the facts and intermediate conclusions used satisfy the conditions of the corresponding rule, and (2) whether the derived conclusion is correct. We then compare all verified correct steps with the gold-standard reasoning steps.

If the model fails to produce the correct final answer, its score is determined by the proportion of intermediate conclusions from the gold-standard reasoning that it correctly derives (i.e., the reasoning steps are verified), regardless of whether it follows the exact same reasoning path. If the model produces a fully correct reasoning chain that leads to the correct final answer, even if its steps differ from the gold-standard sequence, we assign the maximum score.

Process accuracy requires a model not only to produce the correct final answer, but also to present a clear, logically sound, and verifiable reasoning process. This is particularly important in scenarios such as automated theorem proving, multi-step medical diagnosis reasoning, and financial auditing, where the transparency, rigor, and correctness of each step are as critical as the final result. Unlike answer accuracy, which is either 0 or 1, process accuracy can award partial credit based on the correctness of intermediate steps. Thus, even if the final answer is wrong, models demonstrating partially correct reasoning can still receive proportionate scores.

## F.3 TRAINING SETTINGS

We fine-tuned the models using PyTorch 2.x, Transformers and Peft (Wolf et al., 2020), adopting the LoRA configuration of r = 32, lora_alpha = 64, and lora_dropout = 0.05. The target modules included q_proj, k_proj, v_proj, o_proj, gate_proj, up_proj, and down_proj. Unless otherwise specified, the batch size was set to 8. Except for training Llama3.1-8B-instruct on RTX 5880 Ada, all other experiments were conducted on NVIDIA RTX 3090 and RTX 4090 GPUs. For evaluation, we used vLLM (Kwon et al., 2023) with sampling parameters set to temperature = 0.3, top_p = 0.8, and top_k = 20.

**Benchmark-specific adaptation.** For the benchmark-specific adaptation setting (Section 4.2), we fine-tuned each model separately for each downstream benchmark. Hyperparameters, including the learning rate, were tuned independently for every benchmark using the corresponding validation set. The learning rates used in this setting are listed in Table 10.

For Llama3.2-1B-instruct, we utilized the Recall Adam Optimizer (Chen et al., 2020) with the following hyperparameters: $\beta_1 = 0.9$, $\beta_2 = 0.999$, $\epsilon = 1 \times 10^{-8}$, anneal_fun = sigmoid, anneal_k = 0.2, anneal_t0 = 100, anneal_w = 1.0, and pretrain_cof = 3000 for the benchmarks: GSM8k, MATH, MATHQA, SVAMP, MAWPS, AIME, RuleTaker, ProofWriter, LogiQA, ReClor, AbductionR, MMLU, CLUTRR, SLR-Bench and ProntoQA. For all other benchmarks (FOLIO and FLD), the standard Adam Optimizer, with $\beta_1 = 0.9$, $\beta_2 = 0.999$, and $\epsilon = 1 \times 10^{-8}$, was employed.

For Llama3.1-8B-instruct, we retained the same optimizer hyperparameters as for Llama3.2-1B-instruct, but varied which benchmarks used each optimizer. Recall Adam was applied to GSM8K, MATH, MAWPS, AIME, RuleTaker, FLD, LogiQA, SLR-Bench, and standard Adam to the remaining benchmarks: MATHQA, SVAMP, ProofWriter, FOLIO, ReClor, AbductionR, MMLU, ProntoQA, CLUTRR.

For Qwen3-1.7B, the Recall Adam Optimizer was applied to the benchmarks: GSM8k, MATH, MATHQA, SVAMP, MAWPS, AIME, LogiQA, and RuleArena with identical hyperparameters. The standard Adam Optimizer was used for the remaining benchmarks, including RuleTaker, ProofWriter, FOLIO, FLD , AbductionR, MMLU, CLUTRR, SLR-Bench and ProntoQA.

We used different synthesized training datasets depending on the benchmark category and, for Llama3.1-8B-instruct, also treated the training dataset choice as a tunable option. Concretely, for Llama3.2-1B-instruct and Qwen3-1.7B, models evaluated on mathematical reasoning benchmarks were fine-tuned on the synthesized EL-Train dataset (to enhance numerical reasoning skills), whereas models evaluated on logical reasoning benchmarks (including RuleArena) were fine-tuned on the synthesized EN-Train dataset (to enhance logical reasoning skills). For Llama3.1-8B-instruct, we additionally selected between EL-Train and EN-Train on a per-benchmark basis using the corre-

Table 10: Benchmark-specific learning rates applied during fine-tuning on our dataset for evaluation purposes.

| Model | GSM8k | MATH | MATHQA | SVAMP | MAWPS | AIME | RuleTaker | ProofWriter | FOLIO |
|---|---|---|---|---|---|---|---|---|---|
| Llama3.2-1B-instruct | 2e-8 | 8e-9 | 2e-8 | 8e-9 | 8e-9 | 8e-9 | 2e-8 | 5e-7 | 5e-8 |
| Llama3.1-8B-instruct | 2e-8 | 2e-8 | 5e-7 | 2e-7 | 2e-8 | 2e-8 | 2e-8 | 5e-7 | 5e-7 |
| Qwen3-1.7B | 1e-7 | 2e-8 | 5e-8 | 5e-8 | 8e-8 | 8e-8 | 2e-8 | 2e-7 | 2e-7 |

| Model | FLD | LogiQA | ReClor | AbductionR | RuleArena | MMLU | CLUTRR | SLR-Bench | ProntoQA |
|---|---|---|---|---|---|---|---|---|---|
| Llama3.2-1B-instruct | 2e-7 | 5e-7 | 2e-8 | 5e-7 | 5e-8 | 5e-8 | 5e-8 | 2e-7 | 5e-8 |
| Llama3.1-8B-instruct | 5e-8 | 5e-8 | 2e-7 | 5e-7 | 2e-7 | 5e-7 | 5e-7 | 2e-8 | 5e-7 |
| Qwen3-1.7B | 5e-7 | 5e-7 | 5e-7 | 2e-8 | 2e-7 | 2e-7 | 2e-7 | 2e-7 | 5e-8 |

Table 11: Performance of LLMs under varying few-shot settings. 'Proc' refers to process accuracy and 'Ans' refers to answer accuracy.

| Model | #Params. | #Shots | EL-EN | | EL-HN | | HL-EN | | HL-HN | | Average | |
|---|---|---|---|---|---|---|---|---|---|---|---|---|
| | | | Proc | Ans | Proc | Ans | Proc | Ans | Proc | Ans | Proc | Ans |
| Llama3.2-3B-instruct | 3.21B | 0-shot | 4.07 | 33.20 | 0.10 | 11.60 | 0.20 | 14.00 | 0.00 | 3.00 | 1.09 | 15.45 |
| | | 1-shot | 4.30 | 36.60 | 0.47 | 9.20 | 0.20 | 19.80 | 0.00 | 2.80 | 1.24 | 17.10 |
| | | 2-shot | 1.60 | 22.80 | 0.10 | 4.20 | 0.43 | 22.20 | 0.07 | 2.60 | 0.55 | 12.95 |
| | | 3-shot | 1.57 | 27.80 | 0.40 | 4.60 | 0.47 | 20.40 | 0.00 | 4.00 | 0.61 | 14.20 |
| | | 4-shot | 1.50 | 26.60 | 0.10 | 4.40 | 0.51 | 23.20 | 0.00 | 3.80 | 0.53 | 14.50 |
| | | 5-shot | 2.27 | 31.00 | 0.40 | 9.60 | 1.14 | 25.80 | 0.00 | 3.40 | 0.95 | 17.45 |
| Llama3.1-8B-instruct | 8.03B | 0-shot | 20.87 | 50.20 | 5.90 | 16.00 | 2.31 | 34.00 | 0.83 | 5.00 | 7.48 | 26.30 |
| | | 1-shot | 12.33 | 44.60 | 1.27 | 7.40 | 0.18 | 27.80 | 0.13 | 4.80 | 3.48 | 21.15 |
| | | 2-shot | 14.10 | 46.60 | 3.57 | 9.80 | 0.55 | 32.00 | 0.22 | 3.20 | 4.61 | 22.90 |
| | | 3-shot | 12.60 | 48.60 | 2.97 | 13.20 | 0.29 | 34.60 | 0.17 | 5.20 | 4.01 | 25.40 |
| | | 4-shot | 13.13 | 50.60 | 2.80 | 11.20 | 0.15 | 30.80 | 0.37 | 5.00 | 4.11 | 24.40 |
| | | 5-shot | 14.03 | 45.20 | 1.60 | 9.20 | 0.31 | 30.60 | 0.04 | 5.20 | 4.00 | 22.55 |
| Qwen3-8B | 8.19B | 0-shot | 69.30 | 90.80 | 27.80 | 60.80 | 5.05 | 31.40 | 4.48 | 11.40 | 26.66 | 48.60 |
| | | 1-shot | 72.13 | 87.80 | 34.23 | 56.80 | 0.45 | 5.80 | 0.20 | 1.60 | 26.76 | 38.00 |
| | | 2-shot | 78.83 | 90.20 | 62.53 | 81.00 | 0.57 | 13.60 | 0.65 | 4.20 | 35.65 | 47.25 |
| | | 3-shot | 65.67 | 87.40 | 56.37 | 79.80 | 2.55 | 23.00 | 1.64 | 4.80 | 31.56 | 48.75 |
| | | 4-shot | 79.97 | 92.80 | 46.90 | 62.00 | 0.05 | 9.80 | 1.19 | 4.00 | 32.03 | 42.15 |
| | | 5-shot | 80.20 | 92.60 | 68.37 | 81.60 | 0.55 | 12.80 | 1.99 | 6.40 | 37.78 | 48.35 |
| Qwen3-32B | 32.8B | 0-shot | 75.97 | 91.60 | 68.43 | 87.00 | 14.96 | 81.20 | 13.49 | 72.20 | 43.21 | 83.00 |
| | | 1-shot | 80.03 | 86.00 | 71.77 | 81.00 | 16.05 | 83.40 | 13.77 | 73.60 | 45.40 | 81.00 |
| | | 2-shot | 79.13 | 85.20 | 74.23 | 81.40 | 15.76 | 83.20 | 14.77 | 74.20 | 45.97 | 81.00 |
| | | 3-shot | 78.50 | 84.80 | 72.00 | 79.00 | 15.59 | 81.40 | 15.06 | 74.00 | 45.29 | 79.80 |
| | | 4-shot | 71.70 | 78.00 | 75.80 | 82.60 | 15.16 | 83.80 | 17.31 | 78.40 | 44.99 | 80.70 |
| | | 5-shot | 79.10 | 85.00 | 74.27 | 80.00 | 15.74 | 84.20 | 15.48 | 75.20 | 46.15 | 81.10 |
| Phi4 | 14.7B | 0-shot | 65.63 | 89.60 | 57.73 | 76.40 | 2.31 | 36.20 | 2.01 | 20.80 | 31.92 | 55.75 |
| | | 1-shot | 65.10 | 89.80 | 55.13 | 77.20 | 3.02 | 19.80 | 1.43 | 10.60 | 31.17 | 49.35 |
| | | 2-shot | 65.17 | 88.40 | 55.90 | 74.00 | 1.55 | 22.40 | 1.53 | 10.40 | 31.04 | 48.80 |
| | | 3-shot | 63.10 | 90.20 | 51.17 | 74.40 | 2.52 | 36.20 | 2.06 | 10.80 | 29.71 | 52.90 |
| | | 4-shot | 67.50 | 92.20 | 54.50 | 76.40 | 2.39 | 25.80 | 2.12 | 21.80 | 31.63 | 54.05 |
| | | 5-shot | 64.27 | 89.80 | 54.20 | 75.60 | 1.47 | 21.60 | 1.71 | 16.40 | 30.41 | 50.85 |
| Phi4-reasoning-plus | 14.7B | 0-shot | 63.87 | 97.60 | 57.73 | 89.60 | 27.44 | 83.40 | 20.56 | 65.40 | 42.40 | 84.00 |
| | | 1-shot | 61.53 | 96.80 | 55.50 | 89.60 | 28.30 | 83.60 | 20.14 | 65.20 | 41.37 | 83.80 |
| | | 2-shot | 62.50 | 96.80 | 55.13 | 89.60 | 28.05 | 83.20 | 20.39 | 65.40 | 41.52 | 83.75 |
| | | 3-shot | 64.17 | 97.60 | 57.90 | 89.80 | 27.75 | 83.60 | 20.47 | 65.40 | 42.57 | 84.10 |
| | | 4-shot | 62.10 | 97.00 | 55.53 | 89.60 | 28.10 | 83.20 | 20.18 | 65.60 | 41.48 | 83.85 |
| | | 5-shot | 61.50 | 96.80 | 54.90 | 89.60 | 28.79 | 83.20 | 20.28 | 65.40 | 41.37 | 83.75 |

sponding validation set, so that the choice of training dataset was also tuned within the benchmark-specific adaptation setting.

During fine-tuning, we used the validation set provided with each benchmark. If no validation set was available, we randomly sampled a subset from the train split. For the AIME and MAWPS benchmarks, which do not include official validation and train splits, we instead used the test set of AS-Div (Miao et al., 2020) as the proxy validation set. Similarly, for the RuleArena benchmark, which does not include official validation and train splits, we utilized the test set of AR-LSAT (Zhong et al., 2021) as the proxy validation set.

**Unified training on mixed EN-Train and EL-Train.** For the unified training setting (Section 4.2), we trained a single general-purpose model on a mixed synthetic corpus obtained by combining EN-Train and EL-Train. In this setting, we conducted experiments only with Llama3.2-1B-instruct. In this setting, we trained Llama3.2-1B-instruct for 2 epochs with a learning rate of 5e-8 using the Adam Optimizer with the same hyperparameters as in the benchmark-specific adaptation setting.

## G ADDITIONAL EXPERIMENT RESULTS

### G.1 EFFECT OF FEW-SHOT EXAMPLE QUANTITY COMPARED TO ZERO-SHOT

To further investigate how varying the number of in-context examples affects model accuracy, we conducted additional experiments with $k \in \{0, 1, 2, 3, 4, 5\}$ shot settings across 6 selected evaluated models. The results are summarized in Table 11. Consistent with the observations reported in Section 4.1, the performance differences across different $k$ do not exhibit a consistent trend: in some cases, adding examples yields notable improvements, whereas in others it leads to performance

drops. This indicates that, for the highly complex reasoning tasks, increasing the number of in-context examples does not guarantee better performance and may sometimes even be harmful.

## G.2 REPRESENTATIVE CASES

As described in Section 4.1, we conducted a case study. Below, we present representative examples illustrating both the robustness and advantage of process accuracy, as well as three typical types of process errors.

**Robustness and Advantage of Process Accuracy.** Case 1 and Case 2 demonstrate situations where the model's reasoning process followed a different order from the gold-standard reasoning process, yet process accuracy was correctly awarded. Case 3 illustrates that when the model produced the relationship *hampers*, which differs from the gold-standard relationship *hamper* but shares the same lemma, the evaluation correctly handled this case. Case 4 presents an example where the model derived the correct answer through an entirely different reasoning path than the gold-standard one, essentially finding a shortcut. This is possible and considered correct because our synthesizer guarantees logical consistency (i.e., no derived facts, including the query, contradict each other), while allowing the existence of multiple valid reasoning paths. These cases collectively highlight the robustness of our process accuracy evaluation.

Moreover, in Case 2, the model made an error only in the final computation step. As a result, answer accuracy assigned a score of 0, whereas process accuracy awarded $\frac{5}{6}$. Similarly, in Case 5, the model correctly derived an intermediate step that contributed toward the final answer, and process accuracy appropriately gave partial credit. In Case 1, however, the model miscalculated 25913 as 26047, but due to the presence of an expression $\min(88 \times 26047 - 96, -69) = -69$, the final answer happened to be correct. In this case, answer accuracy assigned a score of 1, while process accuracy awarded 0.6, successfully identifying the error. These cases collectively highlight the advantage of process accuracy over answer accuracy. Whereas answer accuracy provides a coarse, binary evaluation, process accuracy offers a more fine-grained and informative assessment: it not only recognizes partially correct reasoning steps but also distinguishes between coincidentally correct answers and genuinely correct reasoning, thus offering a more faithful measure of a model's reasoning ability.

**Cases of Typical Process Errors.** Case 6 and Case 7 illustrate *incorrect application of rules*. In Case 6, the model mistakenly applied an irrelevant fact, while in Case 7 it omitted a relevant fact; nevertheless, both cases still led to the correct intermediate or final conclusion. Case 1 is an example of *incorrect numerical computations*, where the model made an error in an intermediate calculation, but the mistake was masked by a subsequent $\min$ operation, resulting in the correct final answer. Additionally, Case 8 and Case 9 demonstrate *incorrect intermediate results*. In Case 8, the error arose from a lexical mistake in an entity, while in Case 9, it resulted from reversing the direction of a relationship.

---

**Case 1: DeepSeek-R1 in 0-shot setting on HL-HN-4**

```
Process Accuracy = 0.6, Answer Accuracy = 1

Gold-Standard Reasoning Process:
rule_6 & fact_2 & fact_8 => fact_i1: resubmit exists between Arvie
    and Ruperta.
rule_11 & fact_5 & fact_6 & fact_3 & fact_12 => fact_i2: Arvie's
    whole is -71 * -92 - 57 = 6475.
rule_14 & fact_11 & fact_i2 & fact_1 & fact_7 => fact_i3: Arvie's
    imposing is 70 * 99 - 82 = 6848.
rule_13 & fact_10 & fact_14 & fact_15 & fact_i2 => fact_i4: Arvie's
    canadian is subtraction(4 * 6475 - 67, -80) = 25913.
rule_2 & fact_4 & fact_i3 & fact_i1 & fact_i4 & fact_13 => fact_i5:
    Granville's withdrawn is min(88 * 25913 - 96, -69) =
    -69.

Structured Reasoning Process:
```

```
rule_11 & fact_3 & fact_6 & fact_5 & fact_12 => fact_i1: Arvie's
    whole is 6475.
rule_14 & fact_1 & fact_i1 & fact_11 & fact_7 => fact_i2: Arvie's
    imposing is 6848.
rule_6 & fact_8 & fact_2 => fact_i3: resubmit exists between Arvie
    and Ruperta.
rule_13 & fact_15 & fact_14 & fact_10 & fact_i1 => fact_i4: Arvie's
    canadian is 26047.
rule_2 & fact_i2 & fact_i3 & fact_4 & fact_i4 & fact_13 => fact_i5:
    Granville's withdrawn is -69.
```

**Case 2: DeepSeek-R1 in 0-shot setting on HL-HN-8**

**Process Accuracy = $\frac{5}{6}$, Answer Accuracy = 0**

**Gold-Standard Reasoning Process:**
```
rule_14 & fact_18 & fact_1 & fact_9 & fact_16 & fact_2 => fact_i1:
    Suki's acquired is addition(-66 * 53 - 82, 60 * -79 + 67) =
    -8253.
rule_5 & fact_3 & fact_14 & fact_11 => fact_i2: Katrinka's
    impossible is 7 * -52 - 74 = -438.
rule_10 & fact_13 & fact_6 & fact_15 => fact_i3: Marlyn's acquired
    is -65 * 86 + 39 = -5551.
rule_7 & fact_12 & fact_i1 & fact_17 => fact_i4: Suki's executive
    is 80 * 42 - 8 = 3352.
rule_11 & fact_8 & fact_i2 & fact_5 & fact_i3 & fact_7 => fact_i5:
    Jeramie's soaring is subtraction(-33 * -5551 - 12, -28) =
    183199.
rule_6 & fact_i4 & fact_4 & fact_10 & fact_i5 => fact_i6: Suki's
    alternative is -84 * 183199 - 48 = -15388764.
```

**Structured Reasoning Process:**
```
rule_5 & fact_3 & fact_14 & fact_11 => fact_i1: Katrinka's
    impossible is -438.
rule_10 & fact_6 & fact_13 & fact_15 => fact_i2: Marlyn's acquired
    is -5551.
rule_11 & fact_5 & fact_i1 & fact_8 & fact_i2 & fact_7 => fact_i3:
    Jeramie's soaring is 183199.
rule_14 & fact_18 & fact_1 & fact_9 & fact_16 & fact_2 => fact_i4:
    Suki's acquired is -8253.
rule_7 & fact_12 & fact_i4 & fact_17 => fact_i5: Suki's executive
    is 3352.
rule_6 & fact_4 & fact_10 & fact_i5 & fact_i3 => fact_i6: Suki's
    alternative is -15388668.
```

**Case 3: DeepSeek-R1 in 0-shot setting on EL-EN-184**

**Process Accuracy = 1, Answer Accuracy = 1**

**Gold-Standard Reasoning Process:**
```
rule_13 & fact_15 => fact_i1: stack exists between Ed and Claresta.
rule_8 & fact_i1 => fact_i2: hamper exists between Ed and Claresta.
rule_4 & fact_i2 & fact_2 => fact_i3: Ed's reported is 9 * 3 + 9 =
    36.
```

**Structured Reasoning Process:**
```
rule_13 & fact_15 => fact_i1: stack exists between Ed and Claresta.
```

```
rule_8 & fact_i1 => fact_i2: hampers exists between Ed and Claresta
    .
rule_4 & fact_i2 & fact_2 => fact_i3: Ed's reported is 36.
```

### Case 4: DeepSeek-R1 in 0-shot setting on HL-EN-9

**Process Accuracy = 1.0, Answer Accuracy = 1**

**Gold-Standard Reasoning Process:**
```
rule_14 & fact_14 & fact_5 & fact_13 => fact_i1: Charlton's side is
    4 * 6 + 10 = 34.
rule_11 & fact_3 & fact_15 & fact_7 => fact_i2: assume exists
    between Tedie and Charlton.
rule_10 & fact_4 & fact_9 & fact_i1 => fact_i3: Charlton's regional
    is 2 * 34 + 10 = 78.
rule_9 & fact_i3 & fact_8 & fact_i2 => fact_i4: whipsaw exists
    between Astrid and Charlton.
rule_5 & fact_2 & fact_i4 & fact_i1 => fact_i5: Charlton's unusual
    is 34.
```

**Structured Reasoning Process:**
```
rule_14 & fact_14 & fact_5 & fact_13 => fact_i1: Charlton's side is
    34.
rule_5 & fact_2 & fact_9 & fact_i1 => fact_i2: Charlton's unusual
    is 34.
```

### Case 5: Phi4-reasoning-plus in 0-shot setting on EL-EN-489

**Process Accuracy = 0.5, Answer Accuracy = 0**

**Gold-Standard Reasoning Process:**
```
rule_5 & fact_10 => fact_i1: finalize exists between Bengt and
    Maryann.
rule_13 & fact_i1 => fact_i2: Maryann's iranian is 2.
```

**Structured Reasoning Process:**
```
fact_3 => fact_i1: Maryann's annual is 4.
rule_5 & fact_10 => fact_i2: finalize exists between Bengt and
    Maryann.
rule_13 & fact_i2 => fact_i3: Bengt's iranian is 2.
rule_4 & fact_3 => fact_i4: Maryann's iranian is 4.
```

### Case 6: DeepSeek-R1 in 0-shot setting on EL-HN-153

**Process Accuracy = 0.5, Answer Accuracy = 1**

**Gold-Standard Reasoning Process:**
```
rule_5 & fact_4 & fact_7 => fact_i1: Hetty's legal is -95 * 33 - 16
    = -3151.
rule_6 & fact_3 & fact_i1 => fact_i2: Hetty's willing is addition
    (-3151, -46) = -3197.
```

**Structured Reasoning Process:**
```
rule_5 & fact_4 & fact_7 => fact_i1: Hetty's legal is -3151.
rule_6 & fact_2 & fact_i1 => fact_i2: Hetty's willing is -3197.
```

## Case 7: DeepSeek-R1 in 0-shot setting on HL-HN-5

```
Process Accuracy = 0.5, Answer Accuracy = 1

Gold-Standard Reasoning Process:
rule_7 & fact_10 & fact_2 & fact_1 &
    fact_3 => fact_i1: Garey's liquid is max(52, 1) = 52.
rule_3 & fact_8 & fact_4 => fact_i2: scrutinize exists between
    Garey and Aura.
rule_13 & fact_12 & fact_9 => fact_i3: waive exists between Aura
    and Lonnie.
rule_1 & fact_i1 & fact_i2 & fact_i3 & fact_13 => fact_i4: Garey's
    mean is -31 * -17 + 76 = 603.

Structured Reasoning Process:
fact_4 & fact_8 & rule_3 => fact_i1: scrutinize exists between
    Garey and Aura.
fact_12 & fact_9 & rule_13 => fact_i2: waive exists between Aura
    and Lonnie.
fact_10 & fact_1 & fact_2 &
    rule_7 => fact_i3: Garey's liquid is 52.
fact_i1 & fact_i2 & fact_13 & fact_i3 & rule_1 => fact_i4: Garey's
    mean is 603.
```

## Case 8: GPT5-mini in 0-shot setting on EL-HN-162

```
Process Accuracy = 0, Answer Accuracy = 1

Gold-Standard Reasoning Process:
rule_15 & fact_9 & fact_4 => fact_i1: Bradly's accepting is 42 *
    -29 + 86 = -1132.
rule_12 & fact_3 & fact_i1 => fact_i2: Bradly's conditional is -93
    * -1132 - 25 = 105251.

Structured Reasoning Process:
rule_15 & fact_9 & fact_4 => fact_i1: Bradley's accepting is -1132.
rule_12 & fact_3 & fact_i1 => fact_i2: Bradly's conditional is
    105251.
```

## Case 9: GLM4-0520 in 0-shot setting on EL-EN-53

```
Process Accuracy = 0.3, Answer Accuracy = 1

Gold-Standard Reasoning Process:
rule_14 & fact_9 => fact_i1: unsettle exists between Joscelin and
    Clementia.
rule_2 & fact_5 => fact_i2: Joscelin's go is 7.
rule_1 & fact_i1 & fact_i2 => fact_i3: Clementia's first is 7.

Structured Reasoning Process:
fact_9 & rule_9 => fact_i1: unsettle exists between Clementia and
    Joscelin.
rule_2 & fact_9 => fact_i2: Joscelin's go is 7.
fact_i1 & rule_1 & fact_i2 => fact_i3: Clementia's first is 7.
```

