# OpenReview forum: "LogiNumSynth: Synthesizing Joint Logical-Numerical Reasoning Problems for Language Models"
_ICLR.cc/2026/Conference — Submitted to ICLR 2026_

### Official Review · Reviewer_hYkb · 2025-10-23

**Soundness:** 1
**Presentation:** 3
**Contribution:** 1
**Rating:** 0
**Confidence:** 4

**Summary:**

The paper introduces LogiNumSynth, a synthetic data generator designed to create logical and numerical reasoning tasks for language models. Its main goal is to bridge the gap between these two types of reasoning. The framework offers fine-grained control over logical depth and numerical complexity, allowing the generation of tasks across multiple difficulty levels.

The authors present a dataset composed of 4+1 subsets covering easy/hard logical and easy/hard numerical reasoning, along with an additional “extra hard” subset that combines both. They evaluate language models trained on LogiNumSynth to measure their logical and numerical reasoning capabilities. Furthermore, they train models on both downstream benchmarks and their own dataset, demonstrating improved performance on the downstream benchmarks they have trained on.

**Strengths:**

* **Relevance and Motivation:** The focus on combining logical and numerical reasoning is well-motivated. It tackles a structured synthesis approach that could help evaluate or improve models’ reasoning abilities.
* **Control over Task Complexity:** Allowing adjustable difficulty and reasoning depth is a valuable feature for both evaluation and curriculum-based training.
* **Process and Answer Evaluation:** Assessing reasoning steps, not just final outputs, aligns with current trends in verifiable reasoning.

**Weaknesses:**

* **Lack of Clarity in Training Setup:**
  The description of the training and evaluation procedure is inconsistent. While the paper claims *out-of-domain evaluation* after fine-tuning on LogiNumSynth, Appendix F.3 suggests that the models were directly trained on downstream benchmarks (e.g., GSM8K, LogiQA). This critical detail is omitted from the main text, giving the impression that improvements stem from LogiNumSynth training, when in fact they may result from training on the downstream datasets themselves. Table 5 reinforces this concern. The current wording misleads readers about the dataset’s generalization effects and must be corrected.

If this interpretation is incorrect, I invite the authors to clarify. However, if my understanding is accurate, the current presentation seriously misleads readers. This is a major concern and I therefore must strongly recommend rejection unless this issue is convincingly addressed.

* **Unclear Novelty Claim:**
  The claim of being the *first dataset* to integrate logical and numerical reasoning is questionable. Numerical operators are already intrinsic to many logical reasoning tasks [1, 2, 3]. The authors should explicitly compare LogiNumSynth to prior datasets that also integrate logical and numerical reasoning.

* **Limited Training Scope:**
  The paper does not evaluate whether fine-tuning on LogiNumSynth improves or maintains performance on downstream reasoning benchmarks not included at training timesuch as ProntoQA, CLUTRR, or SLR. Such experiments would be essential to demonstrate the dataset’s real-world usefulness beyond its synthetic scope.

* **Missing Rationale for Separate Training:**
  The authors train logical and numerical subsets separately, despite positioning their work as an integration of the two. A joint training setup would be a more natural test of the dataset’s stated purpose and could reveal cross-domain reasoning benefits.

* **Insufficient Explanation of Data Quality:**
  The paper provides little evidence regarding the semantic coherence or plausibility of the synthesized samples. If tasks are generated purely through random sampling, they risk being logically inconsistent or meaningless. Including representative examples and a qualitative analysis of data quality would strengthen the paper’s credibility.

---

**References**

[1] Helff et al., *SLR: Automated Synthesis for Scalable Logical Reasoning*

[2] Ma et al., *KOR-Bench: Benchmarking Language Models on Knowledge-Orthogonal Reasoning Tasks*

[3] Liu et al., *SynLogic: Synthesizing Verifiable Reasoning Data at Scale for Learning Logical Reasoning and Beyond*

**Questions:**

## Revise Empirical results
* Clarify in the main body that the model is also fine-tuned on the downstream dataset.
* Just train your model only on the LogiNumSynth train set (EN-train+EL-train) and check for downstream improvements on the out-of-domain datasets
* If your domain scope is to narrow and the model overfits, you can compare your extensive training run with a model trained only on the downstream benchmarks (excluding LogiNumSynth).
* Also evaluate on MMLU and see whether you can maintain performance or the degradation is not too large, and on inductive logical reasoning to see whether you generalize OOD (e.g., SLR-Bench or CLUTRR)?
* You might also want to compare your tuned model to other logic-tuned models (e.g. 'AIML-TUDA/Llama-3.1-8B-SLR'). If you do so, remeber to also evaluate on OOD dataset, both models have not been trained on.
* Also, add how well the tuned model performs on the test set of LogiNumSynth.

## Add clarifications
* How is the overlap between train/test and within the dataset prevented?
* Specify number of samples of LogiNumSynth datasets in the main body of the paper?
* How is correctness of a solution verified? exact match?
* Also report the selected “reasoning effort” parameter for GPT-5-mini?
* Have you considered using RL-based reasoning training (e.g., GRPO, GSPO, ...) on LogiNumSynth?

While I remain unconvinced by some technical aspects, I find the general idea promising and the problem important. If the authors can address my concerns, I would be open to reconsidering my score.

---

> ### Author Response · Authors · 2025-11-23
> **Author response (1/4)**
>
> We thank the reviewer for their time and comments. There are **several significant misunderstandings** that we would like to clarify.
>
> # Our response to W1:
> > Lack of Clarity in Training Setup: Appendix F.3 suggests that the models were directly trained on downstream benchmarks (e.g., GSM8K, LogiQA).
>
> **This is a significant misunderstanding.** To clarify, **none of our experiments involved training or fine-tuning on any downstream benchmarks** (e.g., GSM8K, LogiQA), **as clearly stated in lines 458 and 1338**. All results in Section 4.2 come from either (1) direct evaluation on these benchmarks, or (2) fine-tuning on our synthesized LogiNumSynth datasets (EL-Train or EN-Train) followed by direct evaluation.
>
> # Our response to W2:
> > Unclear Novelty Claim: The claim of being the first dataset to integrate logical and numerical reasoning is questionable. Numerical operators are already intrinsic to many logical reasoning tasks [1, 2, 3]. The authors should explicitly compare LogiNumSynth to prior datasets that also integrate logical and numerical reasoning.
>
> **This is a significant misunderstanding.** We would like to clarify that **we did not claim anywhere in the paper that LogiNumSynth is the first dataset to integrate logical and numerical reasoning**. Our contribution lies instead in combining (1) joint logical-numerical task synthesis, (2) controllable logical and numerical complexity, and (3) process-level supervision enabled by executable verifiers.
>
> The datasets recommended by the reviewer [1,2,3] do not fall under the category of joint logical-numerical reasoning datasets:
> - SLR [1] focuses on symbolic induction without arithmetic operations or quantitative constraints.
> - KOR-Bench [2] contains separate logical and numerical tasks, but not tasks requiring both types of reasoning jointly.
> - SynLogic [3] primarily synthesizes structural/symbolic puzzles without explicit numerical computation.
>
> **None of these works therefore contradict our positioning, and they target a different problem setting than LogiNumSynth.**
> To support this clarification, we provide a more detailed discussion of these datasets below.
>
> Concretely, SLR [1] focuses on pure symbolic inductive reasoning over a finite domain with human-defined grammars, where "numeric" symbols are not used for arithmetic computation or quantitative constraints, and its evaluation is confined to logical benchmarks (e.g., LogiQA, LogiQA2) rather than numerical reasoning. KOR-Bench [2] targets knowledge-orthogonal reasoning with separate logical and numerical tasks, but does not provide tasks where logical structure and numeric computation must be solved jointly, and it is used solely as an evaluation benchmark. SynLogic [3] synthesizes 35 broad logical task types (e.g., Sudoku-like puzzles) that primarily involve structural/symbolic constraints and largely lack explicit arithmetic or quantitative reasoning.
>
> RuleArena is, to our knowledge and as mentioned in our paper, the only existing dataset that explicitly integrates logical and numerical reasoning. The comparison between RuleArena and our work is already detailed in Section 1 and Appendix C.
>
> # Our response to W3:
> > Limited Training Scope: The paper does not evaluate whether fine-tuning on LogiNumSynth improves or maintains performance on downstream reasoning benchmarks not included at training time such as ProntoQA, CLUTRR, or SLR.
>
> We are not entirely sure we fully understand the reviewer's concern, because **we have already conducted evaluations on a broad range of reasoning benchmarks** that are not part of any LogiNumSynth training split (e.g., GSM8K, AIME, FOLIO, LogiQA, AbductionR, etc.). These experiments directly measure the out-of-domain generalization effect of fine-tuning on LogiNumSynth, and the paper reports that **models trained on LogiNumSynth generally maintain or improve performance across these benchmarks**.
>
> Nonetheless, following the reviewer's suggestion, **we additionally conduct experiments on ProntoQA, CLUTRR and SLR**, which are not included in any LogiNumSynth training set, to further assess out-of-domain generalization. The models are still **trained only on LogiNumSynth (EN-Train)**, and then directly evaluated on these new benchmarks without any additional fine-tuning. As shown in the table below, most results improve consistently, **confirming the out-of-domain generalization benefits of LogiNumSynth**.
>
> | Model | ProntoQA | CLUTRR | SLRBench | SLRBench-partial |
> |---|---|---|---|---|
> | Llama3.2-1B-instruct | 47.00 | 7.63 | 0.00 | 29.60 |
> | Llama3.2-1B-instruct + LogiNumSynth (EN-Train)| 49.60 (+2.60) | 8.78 (+1.15) | 0.00 | 30.30 (+0.70) |
> | Qwen3-1.7B | 94.00 | 13.65 | 7.50 | 25.33  |
> | Qwen3-1.7B + LogiNumSynth (EN-Train)| 97.00 (+3.00) | 18.03 (+4.38) | 7.70 (+0.20) | 27.37 (+2.04) |

---

> ### Author Response · Authors · 2025-11-23
> **Author response (2/4)**
>
> # Our response to W4:
> > Missing Rationale for Separate Training: The authors train logical and numerical subsets separately, despite positioning their work as an integration of the two.
>
> **This is a significant misunderstanding.** We would like to clarify that **we did not train logical and numerical subsets separately.** All of our synthesized training datasets contain logical and numerical reasoning simultaneously, and all fine-tuning experiments are conducted on these constructed logical-numerical tasks. Our work does not involve any separate or decoupled training of logical-only or numerical-only subsets.
>
> # Our response to W5:
> > Insufficient Explanation of Data Quality: The paper provides little evidence regarding the semantic coherence or plausibility of the synthesized samples. If tasks are generated purely through random sampling, they risk being logically inconsistent or meaningless.
>
> **This is a significant misunderstanding.** This concern does not apply to our dataset: **our tasks are not generated through unconstrained random sampling.** As described in Section 2, the synthesis process uses programmatically controlled synthesization with constrained stochasticity to produce formal task specifications. Each synthesized instance is verified for correctness at the formal level before being converted into natural language, ensuring that every task is logically sound and semantically coherent.
>
> To further evaluate the quality of the generated natural language (NL) descriptions, we specifically aim to verify whether the NL descriptions accurately and fluently reflect their underlying formal representations. Instead of relying on human annotators---which can introduce author-side biases in judgment---we adopt an LLM-as-judge protocol using GPT-4o-2024-11-20. For each fact or rule, the model rates the pair consisting of the formal specification and its NL description along two dimensions: Faithfulness and Fluency.
> - **Faithfulness (1-5)**: How accurately the NL text preserves the formal meaning, especially whether the formal specification can be reconstructed from it.
> - **Fluency (1-5)**: Grammar, clarity, and naturalness of the NL text.
>
> We apply this evaluation to 6,012 fact pairs and 6,000 rule pairs. The averaged scores are:
> - fact faithfulness: **4.73**
> - fact fluency: **4.91**
> - rule faithfulness: **4.81**
> - rule fluency: **4.99**
>
> These results indicate that the NL descriptions are both highly faithful to the underlying formal specifications and very fluent. The complete per-instance scores and JSON outputs from the LLM-as-judge evaluation, along with the prompt we used, are provided in the supplementary material under the `/quality_scores` directory.
>
> > Including representative examples and a qualitative analysis of data quality would strengthen the paper's credibility.
>
> A representative example is already shown in Figure 1, and additional samples are provided in the supplementary material under `/data` directory.
>
> # Our response to Q1 and Q2:
> > Clarify in the main body that the model is also fine-tuned on the downstream dataset.
>
> > Just train your model only on the LogiNumSynth train set (EN-train+EL-train) and check for downstream improvements on the out-of-domain datasets.
>
> This misunderstanding has been addressed in our response to W1.
>
> # Our response to Q3:
> > If your domain scope is to narrow and the model overfits, you can compare your extensive training run with a model trained only on the downstream benchmarks (excluding LogiNumSynth).
>
> We appreciate the suggestion. **However, our existing experiments already demonstrate that** LogiNumSynth consistently improves performance across diverse downstream benchmarks, showing its effectiveness without requiring an additional comparison.

---

> ### Author Response · Authors · 2025-11-23
> **Author response (3/4)**
>
> # Our response to Q4:
> > Also evaluate on MMLU and see whether you can maintain performance or the degradation is not too large, and on inductive logical reasoning to see whether you generalize OOD (e.g., SLR-Bench or CLUTRR)?
>
> **We have already evaluated on 14 additional reasoning benchmarks, demonstrating out-of-domain generalization.** For example, on AbductionR, an abductive reasoning benchmark, our models show clear gains (+4.5 for Llama3.2-1B-instruct and +2.2 for Qwen3-1.7B), indicating solid OOD improvements beyond the synthetic domain.
>
> **In addition, we follow the reviewer's suggestion and add experiments on MMLU, SLR-Bench and CLUTRR.** Results are shown in the table below, confirming that **fine-tuning on LogiNumSynth generally maintains or improves performance on these out-of-domain benchmarks as well**.
>
> | Model | MMLU | CLUTRR | SLRBench | SLRBench-partial |
> |---|---|---|---|---|
> | Llama3.2-1B-instruct | 37.95 | 7.63 | 0.00 | 29.60 |
> | Llama3.2-1B-instruct + LogiNumSynth (EN-Train) | 38.03 (+0.08) | 8.78 (+1.15) | 0.00 | 30.30 (+0.70) |
> | Qwen3-1.7B | 64.68 | 13.65 | 7.50 | 25.33  |
> | Qwen3-1.7B + LogiNumSynth (EN-Train)  | 67.28 (+2.60) | 18.03 (+4.38) | 7.70 (+0.20) | 27.37 (+2.04) |
>
> # Our response to Q5:
> > You might also want to compare your tuned model to other logic-tuned models (e.g. 'AIML-TUDA/Llama-3.1-8B-SLR'). If you do so, remeber to also evaluate on OOD dataset, both models have not been trained on.
>
> We are not certain such a comparison is necessary, since the **datasets target different reasoning domains**. **Nonetheless, we conducted this comparison and included OOD evaluations that neither model was trained on.** The results shown in the table below indicate that fine-tuning on **LogiNumSynth yields different improvements compared to SLR** across multiple benchmarks.
>
> |Model|GSM8k|MATH|MATHQA|SVAMP|MAWPS|AIME|RuleTaker|ProofWriter|FOLIO|FLD|LogiQA|ReClor|MMLU|AbductionR|RuleArena|CLUTRR|ProntoQA|
> |---|---|---|---|---|---|---|---|---|---|---|---|---|---|---|---|---|---|
> |Llama3.1-8B-instruct|83.40|40.05|50.90|85.66|93.90|9.93|57.30|27.70|39.90|31.10|30.10*|50.00|63.30*|48.40|10.80|17.55|90.60|
> |AIML-TUDA/Llama-3.1-8B-SLR|30.17 (-53.23)|22.45 (-17.60)|22.06 (-28.84)|69.00 (-16.66)|77.19 (-16.71)|13.30 (+3.37)|56.10 (-1.20)|34.30 (+6.60)|52.21 (+12.31)|41.00 (+9.90)|31.00* (+0.90)|53.60 (+3.60)|66.10* (+2.80)|73.90 (+25.50)|5.86 (-4.94)|15.55 (+2.00)|61.60 (-29.00)|
> |Llama3.1-8B-instruct + LogiNumSynth|83.78 (+0.38)|39.71 (-0.34)|54.81 (+3.91)|87.67 (+2.01)|94.19 (+0.29)|10.14 (+0.21)|58.10 (+0.80)|28.80 (+1.10)|43.84 (+3.94)|42.36 (+11.26)|53.74 (+23.64)|51.80 (+1.80)|64.31 (+1.01)|69.10 (+20.7)|11.42 (+0.62)|13.74 (-3.81)|91.60 (+1.00)|
>
> `*` means results reported in the SLR paper.
>
> # Our response to Q6:
> > Also, add how well the tuned model performs on the test set of LogiNumSynth.
>
> **We do not evaluate on LogiNumSynth after training on it**, as it is not aligned with our goal of assessing how synthetic data improves reasoning ability or serves as an evaluation resource. **Nonetheless, since the reviewer requested it, we include the corresponding results in the table below.**
>
> |model|shot num|EL-EN Ans| EL-EN Proc| EL-HN Ans| EL-HN Proc| HL-EN Ans| HL-EN Proc|HL-HN Ans|HL-HN Proc|exHL-HN Ans|exHL-HN Proc|
> |---|---|---|---|----|----|----|----|----|----|---|---|
> |Qwen3-1.7B|0|76.60|30.30|59.20|6.50|29.80|0.25|15.80|0.26|0.50|0.00|
> |Qwen3-1.7B + LogiNumSynth (EN-Train)|0|77.60 (+1.00)|38.23 (+7.93)|62.80 (+3.60)|4.97 (-1.53)|32.20 (+2.40)|0.05 (-0.20)|11.60 (-4.20)|0.70 (+0.44)|0.50|0.00|
> |Qwen3-1.7B + LogiNumSynth (EL-Train)|0|75.40 (-1.20)|32.63 (+2.33)|61.80 (+2.60)|5.40 (-1.10)|32.40 (+2.60)|0.08 (-0.17)|12.80 (-3.00)|0.04 (-0.22)|0.75 (+0.25)|0.00 |
> |Qwen3-1.7B|3|82.60|44.17|65.20|17.73|41.80|0.20|11.60|0.24|0.50|0.00|
> |Qwen3-1.7B + LogiNumSynth (EN-Train)|3|83.80 (+1.20)|47.23 (+3.06)|65.20|19.73 (+2.00)|43.00 (+1.20)|0.04 (-0.16)|13.20 (+1.60)|0.53 (+0.29)|0.00 (-0.50)|0.00|
> |Qwen3-1.7B + LogiNumSynth (EL-Train)|3|82.00 (-0.60)|47.60 (+3.43)|62.60 (-2.60)|25.13 (+7.40)|42.20 (+0.40)|0.00 (-0.20)|13.40 (+1.80)|0.42 (+0.18)|0.25 (-0.25)|0.00|
> |Llama3.2-1B-instruct|0|13.00|1.83|1.40|0.00|7.40|0.00|0.40|0.00|0.25|0.00|
> |Llama3.2-1B-instruct + LogiNumSynth (EN-Train)|0|18.80 (+5.80)|0.27 (-1.56)|0.40 (-1.00)|0.00 |17.80 (+10.40)|0.24 (+0.24)|0.40 |0.00 |0.25 |0.00 |
> |Llama3.2-1B-instruct + LogiNumSynth (EL-Train)|0|12.00 (-1.00)|1.20 (-0.63)|1.80 (+0.40)|0.20 (+0.20)|9.40 (+2.00)|0.00 |0.40 |0.00 |0.00 (-0.25)|0.00 |
> |Llama3.2-1B-instruct|3|13.80|2.33|1.40|0.10|6.80|0.00|0.60|0.00|0.75|0.03|
> |Llama3.2-1B-instruct + LogiNumSynth (EN-Train)|3|21.40 (+7.60)|0.93 (-1.40)|1.00 (-0.40)|0.10 |19.60 (+12.80)|0.04 (+0.04)|0.60 |0.00 |1.75 (+1.00)|0.30 (+0.27)|
> |Llama3.2-1B-instruct + LogiNumSynth (EL-Train)|3|16.20 (+2.40)|2.67 (+0.34)|0.80 (-0.60)|0.00 (-0.10)|6.20 (-0.60)|0.00 |0.60 |0.00 |0.25 (-0.50)|0.25 (+0.22)|

---

> ### Author Response · Authors · 2025-11-23
> **Author response (4/4)**
>
> # Our response to Q7:
> > How is the overlap between train/test and within the dataset prevented?
>
> Our tasks are sampled from a very large combinatorial space, making train-test **overlap statistically negligible**.
>
> # Our response to Q8:
> > Specify number of samples of LogiNumSynth datasets in the main body of the paper?
>
> The dataset configurations, including sample counts, are presented in **Table 7**. We will make this information explicit in the main text for clarity.
>
> # Our response to Q9:
> > How is correctness of a solution verified? exact match?
>
> As described in **Section 3.3** and **Appendix F.2**, we use two metrics: **answer accuracy** and **process accuracy**.
> - **Answer accuracy** checks whether the **final numerical answer** extracted from the model output (via rule-based and LLM-assisted parsing) **exactly matches** the ground truth.
> - **Process accuracy** measures how well the model's intermediate reasoning aligns with the gold-standard reasoning DAG. It assigns **partial credit based on the proportion of gold-standard intermediate conclusions that the model can logically and correctly derive, even if the final answer is wrong**. At the same time, if a model produces a completely different but fully correct reasoning path, it is awarded full credit, as long as all steps are logically valid.
>
> # Our response to Q10:
> > Also report the selected "reasoning effort" parameter for GPT-5-mini?
>
> We used the default configuration, i.e., medium.
>
> # Our response to Q11:
> > Have you considered using RL-based reasoning training (e.g., GRPO, GSPO, ...) on LogiNumSynth?
>
> We appreciate the suggestion. While our current SFT-based experiments already demonstrate the effectiveness of LogiNumSynth in improving reasoning capabilities, we will consider incorporating RL-based approaches (e.g., GRPO, GSPO) in future work.

---

> ### Author Response · Authors · 2025-11-27
> **A gentle reminder**
>
> Dear Reviewer,
>
> As the discussion period is drawing to a close, we are writing to kindly follow up on our responses. It appeared that some key points in our original manuscript may have been misunderstood, so we have taken great care in our responses to provide clear and thorough clarifications.
>
> We sincerely hope that our responses have now fully addressed your concerns. Should you require any additional information or clarification, we would be very pleased to provide it.
>
> Thank you again for your time and valuable feedback.
>
> Sincerely,
>
> The Authors

---

> ### Comment · Reviewer_hYkb · 2025-11-27
>
> I thank the authors for the detailed and comprehensive rebuttal.
>
> The most critical issue driving my initial score concerned the ambiguity around whether downstream benchmarks were included in training. The authors clarified that this is not the case, which I appreciate. On the positive side, the additional experiments strengthen the scope of the paper, in particular the analysis on data quality (W5) and the comparison with other logic-tuned models (Q5).
>
> However, several concerns remain. While the rebuttal was promising, the authors did not update the paper accordingly nor specify how these changes would be incorporated in the final version. I have updated my score to better reflect the current state of the work, but the following issues still require clarification.
>
> ---
>
> ### Ambiguity Regarding Training on Downstream Benchmarks (W1, W3)
>
> The initial concern for W1 and W3 arises from the appendix statement:
>
> > “For the Qwen model, the Recall Adam Optimizer was applied to the benchmarks: GSM8k, MATH, MATHQA, SVAMP, MAWPS, AIME, LogiQA, and RuleArena with identical hyperparameters. The standard Adam Optimizer was used for the remaining benchmarks, including RuleTaker, ProofWriter, FOLIO, FLD and AbductionR.” (Appx., lines 1333–1336)
>
> As written, this suggests that model weights were optimized directly on the downstream benchmarks themselves. Additionally, Table 10 lists dataset-specific hyperparameters, which further suggests that separate models may have been trained (or hyperparameter-tuned) on the downstream datasets.
>
> In contrast, the main body only states:
>
> > “We fine-tuned two models on the synthesized datasets and directly evaluated them on existing reasoning benchmarks.” (lines 459–460)
>
> Thus, the Appendix and the main body point in different directions. If multiple models were trained or hyperparameters were tuned separately for each benchmark, this must be made explicit in the main paper. Otherwise, Table 5 is misleading, as it currently implies that each row corresponds to a single fixed model evaluated consistently across all downstream benchmarks.
>
> More importantly, training or tuning models individually on each downstream benchmark would be methodologically problematic. Such a setup undermines comparability across datasets and risks introducing benchmark-specific overfitting, which weakens the validity of cross-benchmark conclusions. Standard evaluation practice is to train a model once and report its generalization performance across benchmarks under a fixed configuration. The paper should either clarify that this standard protocol was followed or explicitly justify and motivate any deviation from it.
>
> ---
>
> ### Split Training on EN-Train and EL-Train in Table 5 (W4)
>
> Table 5 reports results for models trained separately on EN-Train and EL-Train. It is unclear why training is split rather than using a single joint model per architecture trained on the combined data. Please clarify the rationale for this design choice. From an application perspective, training a single general-purpose model on the full dataset seems more natural and would improve practical usefulness.
>
> ---
>
> ### Dataset Generation Space
>
> Finally, report the approximate size of the combinatorial space from which problems are sampled.

---

> > ### Author Response · Authors · 2025-11-30
> > **Author response (1/2)**
> >
> > # Response Regarding Paper Updates
> >
> > > While the rebuttal was promising, the authors did not update the paper accordingly nor specify how these changes would be incorporated in the final version.
> >
> > In the rebuttal, **we provided detailed and clear clarifications for all concerns** and **we haved incorporated the promised clarifications and necessary revisions into the new version**.
> >
> > # Clarification on Ambiguity Regarding Training on Downstream Benchmarks (W1, W3)
> > > Thus, the Appendix and the main body point in different directions. If multiple models were trained or hyperparameters were tuned separately for each benchmark, this must be made explicit in the main paper.
> >
> > We did not anticipate that our phrasing would lead to such misunderstanding. **To avoid ambiguity, we have clarified in the main body** like "hyperparameters were tuned separately for each benchmark".
> >
> > # Response on Split Training and Benchmark-Specific Configurations (W1, W3, W4)
> > > More importantly, training or tuning models individually on each downstream benchmark would be methodologically problematic. Such a setup undermines comparability across datasets and risks introducing benchmark-specific overfitting, which weakens the validity of cross-benchmark conclusions. Standard evaluation practice is to train a model once and report its generalization performance across benchmarks under a fixed configuration. The paper should either clarify that this standard protocol was followed or explicitly justify and motivate any deviation from it.
> >
> > > It is unclear why training is split rather than using a single joint model per architecture trained on the combined data. Please clarify the rationale for this design choice. From an application perspective, training a single general-purpose model on the full dataset seems more natural and would improve practical usefulness.
> >
> > We thank the reviewer for the thoughtful comments. While training a single model across all benchmarks is a reasonable setup, **it is not the only valid or practically meaningful one**. **Our split-training design reflects realistic usage patterns and matches the structural characteristics of LogiNumSynth**, where different components emphasize distinct reasoning skills. We have updated the main text to clarify this rationale and to make clear that our configuration and the reviewer's suggestion represent two sound evaluation perspectives, each highlighting different but complementary aspects of model capability. Below, we summarize the key points of our justification:
> >
> > **First, in many real applications, task-/scenario-specific adaptation of smaller models is more practical than deploying one universal large model.** Practitioners often fine-tune a compact backbone for a target domain or benchmark, rather than expecting a single general model to perform optimally everywhere.
> >
> > **Second, this focus follows from the properties of LogiNumSynth.** Our dataset is a controllable synthesis framework, not a fixed distribution: operator sets and composition depth can be instantiated to match different reasoning regimes. Therefore, one key question we study is how benchmark-aligned configurations improve the corresponding tasks, which naturally leads to benchmark-specific tuning experiments.
> >
> > **Third, our training uses LoRA-based fine-tuning.** We do not retrain separate full models; instead, we attach lightweight LoRA adapters to a shared backbone. This mirrors realistic workflows where a small adapter is added for a narrow reasoning domain, making the split setting closer to practical deployment.
> >
> >
> > That said, we fully agree that unified training is also important. Following the reviewer's suggestion, we additionally trained a single general-purpose model on the combined LogiNumSynth data with a fixed configuration and evaluated it across benchmarks. Compared to the benchmark-specific setting, this unified configuration yields clear gains on several reasoning-focused benchmarks such as GSM8k and LogiQA, but also leads to noticeable trade-offs on others, most prominently on the broad-coverage knowledge QA benchmark MMLU. The results are reported below. **We have included both benchmark-specific adaptation and unified training in the new version.**
> >
> > |Model|GSM8k|MATH|MATHQA|SVAMP|MAWPS|AIME|RuleTaker|ProofWriter|FOLIO|FLD|LogiQA|ReClor|AbductionR|RuleArena|MMLU|CLUTRR|SLR-Bench|ProntoQA|
> > |---|---|---|---|---|---|---|---|---|---|---|---|---|---|---|---|---|---|---|
> > |Llama3.2-1B-instruct|35.50|27.33|22.23|61.00|74.72|2.85|46.18|24.84|35.47|32.60|12.85|28.43|47.30|1.96|37.95|7.63|0.00|47.00|
> > |Llama3.2-1B-instruct + LogiNumSynth (unified training)|44.35 (+8.85)|28.06 (+0.73)|25.80 (+3.57)|61.00|75.30 (+0.58)|3.06 (+0.21)|46.60 (+0.42)|24.90 (+0.06)|35.47|31.90 (-0.70)|15.53 (+2.68)|24.90 (-3.53)|48.10 (+0.80)|0.84 (-1.12)|21.05 (-16.90)|5.82 (-1.81)|0.00|51.63 (+4.63)|

---

> > ### Author Response · Authors · 2025-11-30
> > **Author response (2/2)**
> >
> > # Dataset Generation Space
> > We report approximate expressions for the combinatorial space of the dataset generation pipeline for the EL–EN and exHL–HN (depth=10) configurations.
> >
> > The approximate generation space for EL-EN is $10^{245}$. In the following expression, the first three terms correspond to the selection of world elements and queries, while the last two terms correspond to the combinatorial synthesis of facts and rules. The actual synthesis process is more complex, and this serves only as a rough approximation:
> > $$
> > \binom{7944}{10} \cdot \binom{1366}{15} \cdot \binom{976}{10}
> > \times
> > (10 \times 15)
> > \times
> > \text{fact-space}^{15}
> > \times
> > \left[\text{fact-space} \cdot (\text{expr-space} + 10 \cdot 9 \cdot 10)\right]^{15}
> > $$
> > $$
> > \text{fact-space} = 10 \cdot 15 \cdot 10 + 10 \cdot 9 \cdot 10
> > $$
> > $$
> > \text{expr-space} = 10 + 15 + 10\cdot 10\cdot 15
> > $$
> >
> > The approximate generation space for exHL-HN (depth=10) is $10^{3330}$ and the corresponding expression is as follows:
> > $$
> > \binom{7944}{30} \cdot \binom{1366}{40} \cdot \binom{976}{40}
> > \times
> > (30 \times 40)
> > \times
> > \text{fact-space}^{150}
> > \times
> > \left[(\text{fact-space}^3+\text{fact-space}^4+\text{fact-space}^5+\text{fact-space}^6)\cdot
> > (\text{expr-space} + 30\cdot29\cdot40)\right]^{50}
> > $$
> > $$
> > \text{fact-space} = 30 \cdot 40 \cdot 200 + 30 \cdot 29 \cdot 40
> > $$
> > $$
> > \text{sub-expr} = 200 + 40 + 200\cdot 200 \cdot 40
> > $$
> > $$
> > \text{expr-space} = 200\cdot 200 + 4\cdot(\text{sub-expr})^2
> > $$

---

> > ### Author Response · Authors · 2025-11-30
> > **Summary of Responses to Reviewer hYkb**
> >
> > We would like to summarize the evaluation trajectory for Reviewer hYkb.
> >
> > # First Stage
> > This initial rating 0 was based on **a major misunderstanding**---specifically, the assumption that our models were trained on downstream benchmarks. In the rebuttal, we clearly demonstrated that all training is conducted exclusively on LogiNumSynth, and downstream datasets are used only for evaluation. **The reviewer acknowledged this clarification and has since raised the score to a normal level.**
> >
> > # Second Stage
> > After resolving the above misunderstanding, the reviewer raised the score but **maintained one remaining concern**. We explained that **our setup better reflects practical usage**, where downstream tasks are trained with task-specific adapters and different LogiNumSynth subsets correspond to distinct reasoning regimes. To address the reviewer's preferred protocol, **we additionally did an experiment following the protocol**. All requested experiments and clarifications have been incorporated into the revision.
> >
> > # At Last
> > Beyond resolving these central misunderstandings, **we conducted all additional experiments requested by the reviewer**---including evaluations on MMLU, CLUTRR, SLRBench, ProntoQA, unified-model training, comparison with logic-tuned baselines, and data-quality assessments---**and resolved all concerns**. **We have incorporated all corresponding clarifications and revisions into the manuscript.**
> >
> > Given that **the initial 0 score resulted from a misunderstanding that has been fully resolved**, and that **the remaining concern behind the updated score has also been addressed** with additional explanations and experiments, we respectfully ask the AC to consider the paper based on the clarified methodology and strengthened revision.

---

### Official Review · Reviewer_noa9 · 2025-10-25

**Soundness:** 3
**Presentation:** 3
**Contribution:** 3
**Rating:** 6
**Confidence:** 3

**Summary:**

This paper introduces LogiNumSynth, a synthesizer for creating natural language tasks that require combined logical and numerical reasoning, with fine-grained control over difficulty. The tool is used to evaluate and diagnose specific weaknesses in large language models' joint reasoning capabilities. Furthermore, the synthesized data is shown to be effective for targeted training, improving the models' performance on these challenging reasoning tasks.

**Strengths:**

It introduces LogiNumSynth, a novel tool that can generate a wide range of joint logical-numerical reasoning problems with fine-grained control over complexity, moving beyond fixed and limited existing datasets.

The framework is designed not just for evaluation but also for targeted training, demonstrating its value as both a diagnostic tool to identify model weaknesses and a means to improve them.

This paper presents a large-scale evaluation of 29 models, demonstrating that training with synthetically generated data enhances model performance on external reasoning benchmarks.

**Weaknesses:**

The performance improvement from SFT on the synthetic data appears to be marginal (approximately 1%).

Experimental results for SFT on larger-scale models (>=7B) are lacking.

How was the correctness and quality of the synthetic data validated? It is recommended to conduct a small-scale human evaluation.

How diverse is the synthetic data? Is there a trade-off between the diversity of the data and its correctness?

**Questions:**

See the weaknesses

---

> ### Author Response · Authors · 2025-11-23
>
> We thank the reviewer for their time and comments.
>
> # Our response to W1:
> > The performance improvement from SFT on the synthetic data appears to be marginal (approximately 1%).
>
> We **respectfully disagree** that the gains are "marginal". **Several models achieve substantial gains** with SFT on our synthetic data:
>
> - ~4%: Llama3.2-1B-instruct on AbductionR; Qwen3-1.7B on ProofWriter, FLD, and ReClor.
>
> - \>10%: Llama3.2-1B-instruct on LogiQA; Qwen3-1.7B on FOLIO.
>
> These results clearly show that the dataset provides significant benefits. Moreover, **even ~1% improvements indicate that the synthetic data offers useful additional signal for challenging reasoning tasks** (e.g. Llama3.2-1B-instruct gains 1.37% on AIME and Qwen3-1.7B gains 2.12% on RuleArena).
>
> # Our response to W2:
> > Experimental results for SFT on larger-scale models (>=7B) are lacking.
>
> We acknowledge the reviewer's concern regarding the lack of SFT experiments on larger-scale models (>=7B). Due to computational constraints, we were initially unable to include such results. Per the reviewer's request, **we paid for additional computing resources and conducted a new experiment**. Results are shown in the table below and our conclusions remain consistent: **SFT on LogiNumSynth yields unnegligible improvements across various benchmarks, even for larger models**.
>
> |Model|GSM8k|MATH|MATHQA|SVAMP|MAWPS|AIME|RuleTaker|ProofWriter|FOLIO|FLD|LogiQA|ReClor|MMLU|AbductionR|RuleArena|
> |---|---|---|---|---|---|---|---|---|---|---|---|---|---|---|---|
> |Llama3.1-8B-instruct|83.40|40.05|50.90|85.66|93.90|9.93|57.30|27.70|39.90|31.10|30.10*|50.00|63.30*|48.40|10.80|
> |Llama3.1-8B-instruct + LogiNumSynth|83.78 (+0.38)|39.71 (-0.34)|54.81 (+3.91)|87.67 (+2.01)|94.19 (+0.29)|10.14 (+0.21)|58.10 (+0.80)|28.80 (+1.10)|43.84 (+3.94)|42.36 (+11.26)|53.74 (+23.64)|51.80 (+1.80)|64.31 (+1.01)|69.10 (+20.7)|11.42 (+0.62)|
>
> `*` means results reported in the SLR paper [1].
>
> [1] Helff et al., SLR: Automated Synthesis for Scalable Logical Reasoning
>
> # Our response to W3:
> > How was the correctness and quality of the synthetic data validated? It is recommended to conduct a small-scale human evaluation.
>
> This concern does not apply to our dataset: our tasks are not generated through unconstrained random sampling. As described in Section 2, the synthesis process uses programmatically controlled synthesization with constrained stochasticity to produce formal task specifications. **Each synthesized instance is verified for correctness at the formal level before being converted into natural language, ensuring that every task is logically sound and semantically coherent.**
>
> To further evaluate the quality of the generated natural language (NL) descriptions, we specifically aim to verify whether the NL descriptions accurately and fluently reflect their underlying formal representations. Instead of relying on human annotators---which can introduce author-side biases in judgment---we adopt an LLM-as-judge protocol using GPT-4o-2024-11-20. For each fact or rule, the model rates the pair consisting of the formal specification and its NL description along two dimensions: Faithfulness and Fluency.
> - **Faithfulness (1-5)**: How accurately the NL text preserves the formal meaning, especially whether the formal specification can be reconstructed from it.
> - **Fluency (1-5)**: Grammar, clarity, and naturalness of the NL text.
>
> We apply this evaluation to 6,012 fact pairs and 6,000 rule pairs. The averaged scores are:
> - fact faithfulness: **4.73**
> - fact fluency: **4.91**
> - rule faithfulness: **4.81**
> - rule fluency: **4.99**
>
> These results indicate that the NL descriptions are both highly faithful to the underlying formal specifications and very fluent. The complete per-instance scores and JSON outputs from the LLM-as-judge evaluation, along with the prompt we used, are provided in the supplementary material under the `/quality_scores` directory.
>
> # Our response to W4:
> > How diverse is the synthetic data?
>
> **The synthetic data is highly diverse.** As described in **Appendix D.1**, we sample entities, attributes, relationships, and parameters from wide ranges, and vary (1) world richness (scale and density), (2) reasoning depth, and (3) arithmetic difficulty. These controls allow the synthesizer to produce tasks spanning simple worlds to highly interconnected reasoning environments.
>
> > Is there a trade-off between the diversity of the data and its correctness?
>
> **There is no trade-off between diversity and correctness** since the correctness is programmatically verified at the formal level. See Appendix E.1 for more details.

---

> ### Author Response · Authors · 2025-11-27
> **A gentle reminder**
>
> Dear Reviewer,
>
> As the discussion period is drawing to a close, we wanted to kindly follow up on our responses. We sincerely hope they have addressed your comments satisfactorily, and we would be happy to provide any additional information or clarification if necessary.
>
> Thank you very much for your time and consideration.
>
> Best regards,
>
> The Authors

---

### Official Review · Reviewer_ikoH · 2025-10-31

**Soundness:** 2
**Presentation:** 2
**Contribution:** 1
**Rating:** 2
**Confidence:** 3

**Summary:**

This paper introduces LogiNumSynth, a natural language problem synthesizer for logical and numerical reasoning tasks. It demonstrates its usage through the creation of four datasets, based on different combinations of both easy and hard numerical and logical tasks, which are then evaluated on several LLM. It is shown that fine-tuning LLM on the synthetic data can improve its performance on several reasoning benchmarks.

**Strengths:**

Overall, the paper is clearly written and easy to follow. The experiments and analyses appear to be fully documented.

**Weaknesses:**

Unfortunately, the paper lacks innovative ideas, impactful contributions, and analytical depth. It is unclear what purpose the proposed method and analysis can serve meaningfully.

1.  The abstract states that LogiNumSynth should serve as a diagnostic tool and a source of datasets for fine-tuning, but it provides no explanation or insight into why this should significantly improve LLM performance or reduce systematic wrong behavior. The slight to moderate performance gains could be attributed to fine-tuning on a somewhat related domain with respect to the reasoning benchmarks.
2.  The diagnostic possibilities are limited to CoT (process) and answer evaluation on logical and numerical tasks of varying complexity, with no theory on how to explain particular good or bad performance or an indication of how to categorize or assess different failure modes.
3.  There is also no theoretical discussion on why current LLM struggle with numerical and logical reasoning or on the possible limitations of fine-tuning for solving these problems. This makes it difficult to place this work in a scientific context, which is necessary for an impactful publication.

**Questions:**

While this paper, with its proposed methods, is clearly scientifically inspired, it at least needs a discussion of its implicit assumptions, expectations, and potential limitations to function as a stand-alone conference publication.

---

> ### Author Response · Authors · 2025-11-18
>
> We thank the reviewer for their time and comments.
>
> ## Our response to W1:
>
> We thank the reviewer for their attention to the rationale behind the observed performance gains.
>
> > Why this should significantly improve LLM performance or reduce systematic wrong behavior.
>
> LogiNumSynth is explicitly designed to integrate logical and numerical reasoning---requiring models to jointly apply logical rules and numerical computations. Training on such integrated tasks strengthens the model's ability not only in combined reasoning but also in isolated logical or numerical reasoning. This directly accounts for the consistent (albeit moderate) performance gains observed across multiple benchmarks.
>
> > Performance gains could be attributed to fine-tuning on a somewhat related domain with respect to the reasoning benchmarks
>
> We believe this is unlikely. As noted in the paper (lines 72 and 205), our synthesis process is independent of domain knowledge. The synthetic facts carry no real-world meaning; **thus, the improvements can only be attributed to enhanced reasoning capabilities**, rather than the acquisition of any related factual domain knowledge.
>
> ## Our response to W2:
>
> > How to categorize or assess different failure modes.
>
> **Our paper (line 429) already provides an initial taxonomy to categorize failure modes**, which includes: (i) incorrect application of rules, (ii) incorrect numerical computations, and (iii) incorrect intermediate results. Representative examples for each category are available in Appendix G.2.
>
> ## Our response to W3:
>
> > Why current LLM struggle with numerical and logical reasoning.
>
> We agree that the theoretical limitations of LLM reasoning are important, and prior work [1-4] has made significant explorations in this direction. **Our paper addresses a complementary but distinct objective**: to provide a controllable synthetic resource for systematic evaluation and targeted improvement. Rather than seeking a theoretical explanation, we deliver a practical and extensible tool designed to empower the community to investigate these very questions more effectively.
>
> [1] Liu et al. Concise and Organized Perception Facilitates Reasoning in Large Language Models
>
> [2] Polyakov et al. Interpretability Analysis of Arithmetic In-Context Learning in Large Language Models
>
> [3] Xu et al. Principled Understanding of Generalization for Generative Transformer Models in Arithmetic Reasoning Tasks
>
> [4] Zheng et al. The Curse of CoT: On the Limitations of Chain-of-Thought in In-Context Learning
>
> ## Our response to Q1:
>
> > Needs a discussion of its implicit assumptions, expectations, and potential limitations.
>
> We thank the reviewer for raising these important points. While these aspects are discussed throughout the paper (e.g., expectations in lines 51–53, limitations in Appendix B), we are pleased to summarize them here for clarity:
>
> Implicit Assumptions: Our framework rests on two key assumptions: (1) that joint logical-numerical reasoning can be effectively modeled using rule-based logical structures combined with numerical expressions, and (2) that expressing these reasoning tasks in natural language preserves their core computational challenges. We further assume that such integrated tasks capture a meaningful subset of real-world reasoning difficulties encountered by LLMs.
>
> Expectations: As outlined in our motivation (lines 51--53), LogiNumSynth was developed with two primary expectations: (1) that its controllable joint reasoning tasks would systematically expose LLM weaknesses in intermediate reasoning steps; and (2) that fine-tuning on these tasks would lead to improvements not only on joint reasoning benchmarks, but also on isolated logical or numerical tasks. Our empirical results provide support for both of these expectations.
>
> Limitations: In addition to those noted in Appendix B, we acknowledge that LogiNumSynth only covers a subset of real-world reasoning scenarios and does not incorporate tasks requiring extensive domain knowledge. Moreover, while synthetic fine-tuning enhances performance, it does not resolve the fundamental theoretical limitations of current LLMs---an important direction for future research that falls outside the scope of this work.

---

> > ### Comment · Reviewer_ikoH · 2025-11-28
> >
> > I thank the authors for their clarifications and additional discussion!
> >
> > W1: This is no explanation, only an intention and an observation. It is also unclear to me how the performance gains can be attributed to the proposed method without conducting ablation testing on other fine-tuning datasets. There is still no specific hypothesis or explanation clarifying the benefits beyond simply fine-tuning on a dataset with mathematical tasks.
> >
> > W2: You provide a taxonomy of categories, but not a method for categorizing or evaluating them. Second, the diagnostic limitation to CoT remains.
> >
> > W3: Thank you for addressing the implicit assumptions! Please incorporate this discussion into the revised paper.
> >
> > Please provide a revised manuscript that incorporates all clarifications, additional results, and discussions of assumptions highlighted by the reviewers in different colors. I am willing to raise my score if the new revision is significantly improved.

---

> > > ### Author Response · Authors · 2025-11-30
> > > **Official Comment by Authors**
> > >
> > > We thank the reviewer for the continued engagement. **The reviewer's concerns primarily reflect a preference for theoretical analysis of LLM reasoning, which is outside the intended scope of our work**---LogiNumSynth is designed as a practical synthetic data generator and evaluation tool, not a theoretical framework.
> > >
> > > Nevertheless, we have fully incorporated all requested clarifications, additional discussions (including assumptions, expectations, and limitations), and updates into a revised manuscript, with all changes clearly highlighted in color as requested.
> > >
> > > The reviewer **explicitly stated willingness to raise the score** upon receiving a significantly improved revision, and we believe the updated manuscript fully addresses the raised points.

---

> ### Author Response · Authors · 2025-11-27
> **A gentle reminder**
>
> Dear Reviewer,
>
> As the discussion period is drawing to a close, we wanted to kindly follow up on our responses. We sincerely hope they have addressed your comments satisfactorily, and we would be happy to provide any additional information or clarification if necessary.
>
> Thank you very much for your time and consideration.
>
> Best regards,
>
> The Authors

---

### Meta-Review · Area_Chair_Dsgg · 2026-01-07

**Summary:**

This paper focuses on LMs’ joint logical-numerical reasoning and presents LogiNumSynth, which allows fine-grained control over reasoning world richness, logical reasoning depth, and the complexity of numerical computations, enabling flexible data synthesis across different difficulty levels. Experiments with multiple LLMs highlight persistent weaknesses in logical-numerical reasoning. Overall, the paper is well presented; however, the insights into logical-numerical reasoning could be deepened, LogiNumSynth requires more thorough analysis, and the training setup remains somewhat controversial.

**Reviewer Concerns:**

During the rebuttal, the authors clarified the motivation for joint logical-numerical reasoning, the quality and diversity of LogiNumSynth, some limited improvements, and experimental setup issues, such as concerns about clarity in the training setup, particularly the fine-tuning details on LogiNumSynth. This work would benefit from a clearer description of the training setup and a well-justified discussion of the motivation and underlying assumptions.

**Reviewer Scores:**

The rebuttal clarifications may warrant a slight score increase, but they are not sufficient to change the acceptance decision.

---

### Decision · Program_Chairs · 2026-01-26

Reject